# InfraRL: A Benchmark for Constrained Resource Allocation in Large-Scale Infrastructure Asset Management

Yantian Wang [1]   Wenhao Li [1]   Bo Jin [1,2]

## Abstract

Optimizing maintenance strategies for large-scale infrastructure is a critical sequential decision-making problem.While Reinforcement Learning (RL) offers a theoretical framework for such problems, practical deployment necessitates offline constrained RL—learning policies solely from static historical datasets under rigid budgetary limits without dangerous on-policy exploration. However, current research is hindered by benchmarks that fail to capture the confluence of distributional shift and hard constraints typical of real-world assets. We introduce InfraRL, a high-fidelity benchmark that uses bridge maintenance as a rigorous testbed for general infrastructure asset management challenges. Constructed from the U.S. National Bridge Inventory, InfraRL defines a rigorous offline task for optimizing maintenance strategies under hard budgetary constraints. We benchmark a diverse suite of baselines, ranging from industry-standard heuristics to single-agent, multi-agent, planning-based, and constrained offline RL methods. Through a comprehensive evaluation protocol, we analyze performance across structural utility, constraint adherence, and behavioral fidelity, revealing critical trade-offs between safety and long-term efficiency. Our code and data are available at https://github.com/BriSky-2021/InfraRL.

## 1. Introduction

Infrastructure Asset Management (IAM) serves as the backbone of modern society, ensuring the reliability of critical networks ranging from power grids to transportation arteries (Wu et al., 2021). As these large-scale systems age globally, decision-makers face a complex sequential decision-making problem: how to optimally allocate limited budgetary resources for inspection and repair while strictly adhering to safety standards (Orcesi & Frangopol, 2011; Kim & Frangopol, 2018). Within this broad domain, highway bridges represent a particularly high-stakes exemplar, embodying the quintessential challenges of IAM—long lifecycles, expensive interventions, and zero tolerance for structural failure. The consequences of suboptimal strategies are starkly illustrated by the 2018 collapse of the Morandi Bridge in Genoa, Italy. This tragedy, attributed to delayed retrofitting, resulted in 43 fatalities and severed a vital economic corridor, highlighting the critical need for intelligent management systems (Calvi et al., 2019). While traditional approaches rely on heuristic rules, Reinforcement Learning (RL) has emerged as a transformative paradigm, offering the potential to automate these high-dimensional decisions directly from data (You et al., 2019).

However, translating the theoretical promise of RL into viable IAM strategies necessitates navigating a confluence of rigorous constraints rarely encountered in standard control tasks. First, the safety-critical nature of infrastructure precludes on-policy exploration. Agents cannot learn by trial-and-error where failure implies structural collapse. This mandates an *offline RL* paradigm, where policies must be derived entirely from static, historical datasets without interacting with the environment (Levine et al., 2020). Second, unlike unconstrained robotic control, IAM is fundamentally a constrained Markov decision process (CMDP) (Altman, 2021). Agents must maximize long-term structural health while strictly adhering to hard, renewable budgetary limits, where violations are operationally inadmissible rather than merely penalized. Furthermore, unlike synthetic environments governed by deterministic physics engines, real-world administrative data introduces significant stochasticity, non-stationarity, and noise, complicating the extraction of reliable reward signals and transition dynamics.

Despite these distinct requirements, current RL benchmarks fail to provide a unified testbed that captures this complexity, as shown in Table 1. General-purpose offline benchmarks, such as D4RL (Fu et al., 2020), typically lack native

---

[1]School of Computer Science and Technology, Tongji University, China [2]Shanghai Research Institute for Intelligent Autonomous Systems. Correspondence to: Wenhao Li <whli@tongji.edu.cn>, Bo Jin <bjin@tongji.edu.cn>.

*Proceedings of the 43rd International Conference on Machine Learning*, Seoul, South Korea. PMLR 306, 2026. Copyright 2026 by the author(s).

cost signals required for constrained optimization. Conversely, safety-oriented environments like Safety-Gym (Ray et al., 2019) or domain-specific simulators like IMP-MARL (Leroy et al., 2023) rely on online interaction or generative deterioration models, bypassing the challenges of learning from finite, noisy historical logs. Consequently, there exists a critical gap: the lack of a high-fidelity benchmark that simultaneously addresses offline learning, hard constraints, and real-world data distributions.

To bridge this divide, we introduce **InfraRL**, a high-fidelity benchmark grounded in the U.S. National Bridge Inventory (NBI), designed to rigorously evaluate offline constrained policy optimization in high-stakes environments. Unlike synthetic tasks, InfraRL addresses the noise inherent in real-world administrative data through a strict *causal verification pipeline*, ensuring that state transitions and reward signals reflect ground-truth maintenance causality rather than recording errors. Furthermore, distinct from standard constrained RL benchmarks that rely on soft penalties, InfraRL enforces operational realism through a *global hard-constraint protocol*. We implement a centralized allocation mechanism that compels agents to maximize network-wide utility under rigid fiscal caps, precluding the violation-tolerant behaviors often learned by unconstrained baselines. Finally, we benchmark a diverse suite of algorithms—ranging from single- to multi-agent policy optimizer and industry-standard heuristics—evaluating them across structural utility, budget adherence, and long-term generalization.

Through extensive empirical evaluation on InfraRL, we highlight several phenomena in applying RL to infrastructure management. Our key findings include:

- **Constraint Internalization amidst Sparsity:** We reveal that specialized constrained agents effectively internalize budgetary limits (achieving a $1.7\%$ raw violation rate). We observe that high behavioral similarity to human priors is consistent with the domain's extreme action sparsity, indicating that performance gains stem primarily from the precise selection of rare, high-impact maintenance interventions rather than broad policy deviations.
- **Breaking the Imitation Ceiling:** We demonstrate that pure imitation learning is strictly bound by historical suboptimality. Unlike imitation agents that plateau regardless of resources, value-based methods exhibit *economic scalability*—successfully translating a $4\times$ budget increase into linear health gains by capturing underlying causal dynamics.
- **Hybrid Superiority for Generalization:** We find that incorporating domain heuristics into RL training acts as a vital stabilizer against distribution shift. This hybrid approach prevents training collapse and enables superior generalization in long-horizon (100-year) simulations compared to pure RL or pure rule-based methods.

**Conflict of Interest Disclosure.** The authors declare no financial conflict of interest relevant to this work.

## 2. Problem Formulation

We characterize the bridge maintenance problem as a CMDP with a factorized state space and instantaneous hard constraints, defined by the tuple $\mathcal{M} = \langle \mathcal{S}, \mathcal{A}, P, R, C, \gamma, B \rangle$. The system consists of $N$ agents (bridges), where the global state space $\mathcal{S}$ and action space $\mathcal{A}$ are the Cartesian products of the individual bridge spaces, denoted as $\mathcal{S} = \bigtimes_{i=1}^{N} \mathcal{S}^i$ and $\mathcal{A} = \bigtimes_{i=1}^{N} \mathcal{A}^i$, respectively. At decision epoch $t$ (corresponding to one year between consecutive NBI inspection records), the global state is a vector $\mathbf{s}_t = [s_t^1, \ldots, s_t^N]^\top$, where each local state $s_t^i = (h_t^i, \omega_t^i) \in \mathcal{S}^i$ comprises the structural condition rating $h_t^i \in \mathbb{R}$ and auxiliary features $\omega_t^i$ (e.g., age, traffic, dimensions). The joint action $\mathbf{a}_t = [a_t^1, \ldots, a_t^N]^\top$ is drawn from discrete local action sets $\mathcal{A}^i = \{0, 1, 2, 3\}$, where $a = 0$ denotes No Action, $a = 1$ Minor Repair, $a = 2$ Major Repair, and $a = 3$ Replacement.

The system dynamics are governed by the transition kernel $P : \mathcal{S} \times \mathcal{A} \times \mathcal{S} \rightarrow [0, 1]$. Assuming conditional independence of structural degradation among bridges, the global transition probability factorizes as $P(\mathbf{s}_{t+1} \mid \mathbf{s}_t, \mathbf{a}_t) = \prod_{i=1}^{N} P^i(s_{t+1}^i \mid s_t^i, a_t^i)$, where $P^i$ encodes the stochastic deterioration and maintenance effects derived from empirical data. The cost structure is defined by the function $C(\mathbf{s}, \mathbf{a}) = \sum_{i=1}^{N} c(s^i, a^i)$, representing the total monetary expenditure. Unlike standard CMDPs which often employ cumulative cost constraints, this system is subject to a hard instantaneous budget constraint $B$. This restricts the feasible control space at each timestep; strictly speaking, we define the *admissible action set* $\mathcal{A}_B(\mathbf{s}) \subseteq \mathcal{A}$ for any state $\mathbf{s}$ as $\mathcal{A}_B(\mathbf{s}) = \{\mathbf{a} \in \mathcal{A} \mid C(\mathbf{s}, \mathbf{a}) \leq B\}$.

The reward function is engineered to balance immediate structural improvement with cost efficiency, augmented by potential-based reward shaping (PBRS) (Ng et al., 1999) to ensure policy invariance and long-term health preservation. The total realized global reward $r_t$ at time $t$ is the sum of individual rewards, $r_t = \sum_{i=1}^{N} r^i(s_t^i, a_t^i, s_{t+1}^i)$. Each individual reward component is defined as:

$$r^i(s_t^i, a_t^i, s_{t+1}^i) = \underbrace{(h_{t+1}^i - h_t^i)}_{\Delta \text{Health}} - \beta \underbrace{\left[ \frac{\log(1 + c(s_t^i, a_t^i))}{\log(1 + C_{95\%})} \right]}_{C_{\text{norm}}} + \underbrace{\tau(\gamma h_{t+1}^i - h_t^i)}_{\text{PBRS}},$$

(1)

where $C_{\text{norm}}$ utilizes a log-normalization scaled by the 95th percentile of historical costs ($C_{95\%}$) and a weight $\beta$. The PBRS term is derived from the potential function $\Phi(s^i) = \tau h^i$, satisfying the form $F = \gamma \Phi(s_{t+1}) - \Phi(s_t)$,

*Table 1.* Comparison of offline and constrained RL datasets or benchmarks.

| Feature / Dataset | InfraRL (Ours) | IMP-MARL | SustainGym | DSRL | Safety-Gym | D4RL |
|---|---|---|---|---|---|---|
| Offline Dataset | ✔ | ✗ | ✗ | ✔ | ✗ | ✔ |
| Multi-Agent Support | ✔ | ✔ | ✔ | ✗ | ✗ | ✗ |
| Real-World Data | ✔ | ✗ | ✔ | Hybrid | ✗ | ✗ |
| Native Constraint Labels | ✔ | ✔ | ✔ | ✔ | ✔ | ✗ |

which preserves the optimal policy structure. Specific values for the weighting coefficients and normalization constants are detailed in Appendix C.3. The expected reward function used for optimization is $R(\mathbf{s}, \mathbf{a}) = \mathbb{E}_{\mathbf{s}' \sim P(\cdot|\mathbf{s},\mathbf{a})}[r(\mathbf{s}, \mathbf{a}, \mathbf{s}')]$. The objective is to find a policy $\pi : \mathcal{S} \to \Delta(\mathcal{A})$ that maximizes the expected discounted cumulative reward while ensuring that the budget constraint is never violated. Formally, the optimization problem is:

$$\pi^* = \arg\max_{\pi} \mathbb{E}_{\substack{\mathbf{a}_t \sim \pi(\cdot|\mathbf{s}_t) \\ \mathbf{s}_{t+1} \sim P(\cdot|\mathbf{s}_t,\mathbf{a}_t)}} \left[ \sum_{t=0}^{\infty} \gamma^t R(\mathbf{s}_t, \mathbf{a}_t) \right], \quad (2)$$

subject to the support constraint $\mathrm{supp}(\pi(\cdot|\mathbf{s})) \subseteq \mathcal{A}_B(\mathbf{s})$ for all state $\mathbf{s} \in \mathcal{S}$.

## 3. Benchmark Building

A core contribution of InfraRL is turning the U.S. National Bridge Inventory (NBI)—a longitudinal database maintained by the Federal Highway Administration (FHWA) that records annual inspection and improvement information for public road bridges nationwide—into a high-fidelity offline RL dataset. While the NBI offers unmatched coverage and temporal depth, it is an administrative archive rather than an RL-ready log: key fields (e.g., work codes and cost entries) can be missing, noisy, or weakly aligned with true physical interventions. For example, some records report non-zero expenditures without corresponding maintenance codes, whereas others list maintenance codes without observable improvements in subsequent condition ratings. To address this mismatch between *recorded actions* and *executed interventions*, we apply a three-stage processing and causal verification pipeline (detailed in Appendix C) to produce consistent state–action–reward trajectories, summarized in Figure 1.

### 3.1. Data Processing and Verification Pipeline

**Dataset scope.** We start from the annual NBI public releases spanning 1992–2023 and retain only highway bridges (SERVICE_ON_042A=1) located in California. After cleaning and enforcing minimum-history requirements (§C.1), we obtain longitudinal bridge-level trajectories from which we construct the benchmark via regional sampling (400 regions) and temporal sliding windows (window size 15 years, stride 5 years), yielding 2,000 multi-agent episodes for offline training and evaluation.

**1. Action typing and execution verification.** Raw NBI logs do not reliably indicate whether a maintenance action was truly executed. We therefore adopt a two-stage identification procedure.

*(i) Type candidate from work codes.* We first map WORK_PROPOSED_075A (NBI Item 75A: the recorded work-proposed code indicating the intervention type) into a coarse action *type candidate* (0: No Action, 1: Minor Repair, 2: Major Repair, 3: Replacement) using a fixed dictionary. Notably, negligible-cost activities (e.g., Code 33 "Deck Widening", average cost $\approx$ \$33) are merged into No Action to reduce noise, while structurally significant interventions are grouped into Minor/Major Repairs and Replacement based on cost clustering (see Appendix C.1 for full mapping).

*(ii) Execution validation from physical evidence.* A non-zero candidate is retained as an executed maintenance action only if it coincides with observable evidence of structural intervention, including (a) an improvement in any structural condition rating and/or (b) an update of the reconstruction year. Otherwise, the sample is treated as No Action. This causal verification prevents "phantom" interventions that exist only as administrative entries.

**2. Causality and logical consistency filters.** We remove samples that violate physical/economic causality, including: (1) *Phantom Improvement*: rating increases without any logged maintenance evidence; (2) *Ineffective Repair*: maintenance logged but the subsequent-year rating decreases; (3) *Phantom Cost*: positive expenditures while labeled as No Action; (4) *Administrative Redundancy*: repeated carry-over sequences of identical actions/costs that yield no health improvement. Missing inspection years are imputed using forward-fill to preserve trajectory continuity.

**3. Regional partitioning via neighborhood-based sampling.** To reflect how bridge maintenance is budgeted and executed in practice, we group bridges into geographically coherent *regions* instead of treating each bridge as an independent episode. Intuitively, if each bridge were optimized in isolation, the hard budget constraint would decouple into per-bridge limits and the problem would miss the core difficulty of IAM: *competing* for a shared, finite annual budget where funding one bridge necessarily reduces funds available for others. Regional grouping therefore induces realistic cross-asset coupling (shared budget and local plan-

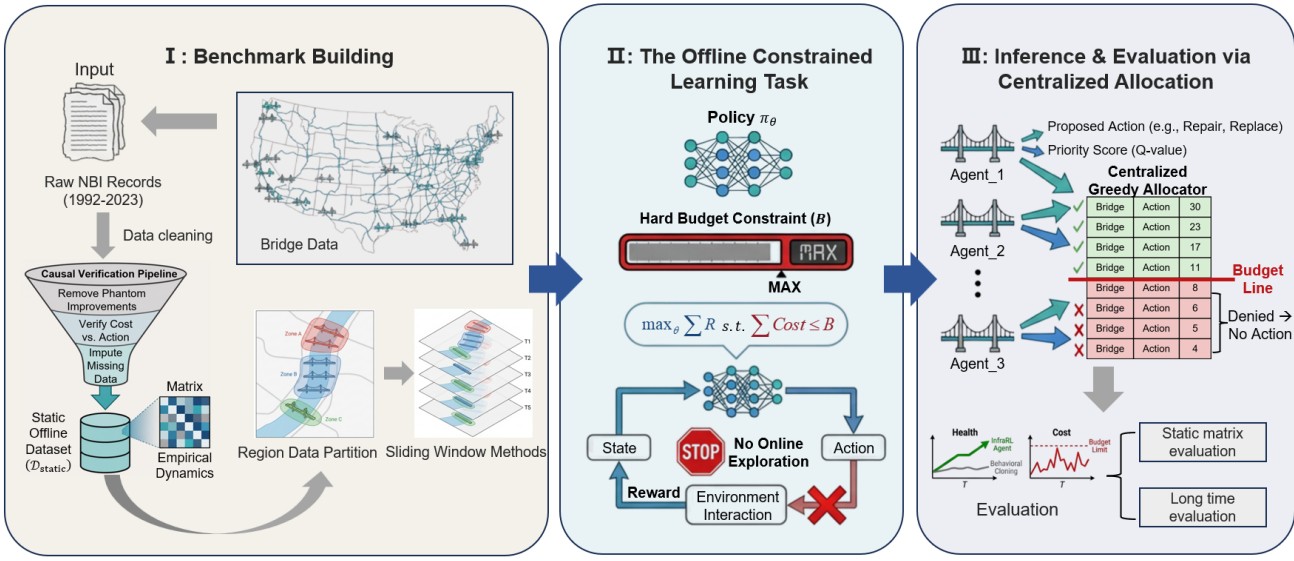

*Figure 1.* Overview of the InfraRL framework. **(I) Benchmark Building:** We transform raw NBI records into a high-fidelity offline RL dataset through a three-stage causal verification pipeline that filters phantom interventions and imputes missing data. Bridges are partitioned into geographically coherent regions to create multi-agent environments with shared budget constraints. **(II) Offline Constrained Learning:** The objective is to learn policies entirely from static datasets that maximize bridge health under hard budget caps, without online exploration. **(III) Inference & Evaluation:** During testing, a centralized greedy allocator ranks agent action requests by priority (e.g., Q-values) and enforces the hard budget limit $B_{r,t}^{\mathrm{eval}}$, ensuring that the executed joint action is always feasible.

ning), yields tractable multi-agent episodes, and enables evaluation of constraint-handling mechanisms (e.g., centralized allocation) under meaningful resource contention. We form regions using a neighborhood-based procedure. We randomly select a bridge as a seed, then include the $N-1$ nearest bridges by Manhattan distance in latitude–longitude. We repeat this process to obtain 400 regions. Each region constitutes an episode with $N$ bridges sharing a regional budget. For each region, we additionally construct a static connectivity matrix based on geographic proximity (Appendix C). Temporal windowing and resulting episode counts are described in the **Dataset scope** paragraph above.

### 3.2. Budget Definition

Since the NBI does not explicitly report region-level annual budgets, we operationalize a per-region, per-year hard cap using a behavioral proxy. We first train a Multi-Agent Behavior Cloning (BC) policy $\pi_{\mathrm{BC}}$ on the offline dataset. For each region $r$ and year $t$, we feed the full set of bridge states $\mathbf{s}_{r,t}$ into $\pi_{\mathrm{BC}}$ to obtain a simulated joint action $\mathbf{a}_{r,t}^{\mathrm{BC}}$, then compute its implied spending

$$C(\mathbf{s}_{r,t}, \mathbf{a}_{r,t}^{\mathrm{BC}}) = \sum_i c(s_{r,t}^i, a_{r,t}^{\mathrm{BC},i}). \tag{3}$$

We define the evaluation budget as a $20\%$ contingency margin over this behavioral estimate:

$$B_{r,t}^{\mathrm{eval}} = 1.2 \times C(\mathbf{s}_{r,t}, \mathbf{a}_{r,t}^{\mathrm{BC}}). \tag{4}$$

We adopt this BC-induced budget (instead of directly summing realized historical costs) to obtain a policy-agnostic, counterfactual budget proxy that (i) is defined purely from observed states, (ii) is less sensitive to the noise/inconsistencies of administrative cost records, and (iii) avoids coupling the test-time constraint to the same action/cost labels used to train and evaluate the agents.

**Inference-Time Budget Allocator.** Offline policies may output a raw joint action $\mathbf{a}_{r,t}^{\mathrm{raw}}$ whose total cost exceeds the hard evaluation budget $B_{r,t}^{\mathrm{eval}}$. To enforce feasibility while preserving the policy's preference ordering, we apply a deterministic centralized greedy allocator at inference time. For each bridge $i$ requesting a non-zero action, we assign a priority score $S_{r,t}^i$: for value-based methods, $S_{r,t}^i = Q(s_{r,t}^i, a_{r,t}^{\mathrm{raw},i})$; for policy-based methods, $S_{r,t}^i$ is the logit/probability of the chosen action. We sort all requested actions by $S_{r,t}^i$ in descending order and approve them greedily as long as the remaining budget can afford their costs; otherwise the corresponding bridge is forced to take No Action. This post-processing guarantees that the executed joint action $\mathbf{a}_{r,t}$ always satisfies $\sum_i c(s_{r,t}^i, a_{r,t}^i) \le B_{r,t}^{\mathrm{eval}}$. Further implementation details are provided in Appendix B.3.

## 4. Evaluation Protocols

InfraRL is derived from static administrative logs rather than a controllable simulator; consequently, evaluation must reflect the *offline* regime in which policies are learned from finite, biased data without on-policy exploration. Our proto-

col is designed to stress the two defining failure modes of real-world IAM: (i) distributional shift induced by offline learning, and (ii) *hard, renewable* budget constraints that couple decisions across many assets. To ensure comparability under operationally admissible spending, we report results under the same inference-time budget enforcement mechanism (Section 3.2). Algorithmic details are provided in Appendix B, and full hyperparameter/computational settings are documented in Appendix E.

## 4.1. Algorithm Overview

To instantiate the above protocol, we benchmark methods that span the spectrum of *offline robustness* and *constraint awareness* under a shared regional budget. Accordingly, we group baselines by (i) how they model cross-asset coordination (centralized vs. decentralized) and (ii) whether constraint handling is learned explicitly or delegated to post-hoc enforcement. Table 2 provides a systematic comparison of their key features.

**Centralized (single-agent) baselines.** We first include methods that treat the regional decision as a centralized control problem and (implicitly or explicitly) reason about the budget. This group includes the constrained value-based MultiTask-CPQ (Xu et al., 2022), the sequence-modeling constrained imitation method CDT (Liu et al., 2023b), and a budget-conditioned imitation baseline MultiTask-BC. These algorithms provide a strong reference point for what is achievable when global optimization is attempted with shared parameters.

**Planning baselines.** We additionally evaluate **MPC+Forecasting**, a fair learned planning baseline that uses a learned one-step health predictor for short-horizon rollouts, and **MPC-Oracle**, a stronger informed reference that uses privileged evaluator-aligned reward information while still advancing imagined states with the learned predictor.

**Decentralized (multi-agent) baselines.** To mirror the operational reality where each bridge is acted on locally and coordination happens only through a shared budget, we also evaluate multi-agent policies. We include two anchor points: Random as a lower bound and Discrete-BC (behavior cloning) as an expert-imitation baseline.

**Value-based offline RL (independent learners).** To isolate the effect of value learning from explicit coordination, we evaluate per-agent value-based offline RL where each agent learns from its local trajectory. We use CQL (Conservative Q-Learning) (Kumar et al., 2020) to mitigate value overestimation under distribution shift, and OneStep RL (Brand-fonbrener et al., 2021) as a one-step policy improvement method over the behavioral policy.

**Offline MARL with centralized training.** Finally, we in-

clude state-of-the-art offline MARL variants that attempt to learn coordinated value estimates under CTDE. In particular, QMIX-CQL augments the QMIX value decomposition architecture (Rashid et al., 2018) with CQL-style conservatism to better handle offline extrapolation, QMIX-CQL-MF replaces the full joint state with compact mean-field group statistics for variable-size regions, and IQL-CQL-MARL represents independent learning enhanced with conservative regularization.

**Heuristic Baseline (Matrix-Aligned).** To contextualize learned performance against a strong, domain-plausible upper reference, we implement a deterministic rule-based policy calibrated to the empirical transition matrix used by the evaluation environment.

- **Action Rules:** choose the locally most improving intervention by condition band: Critical (Rating $\leq 3$) $\rightarrow$ Minor Repair; Poor ($3 <$ Rating $\leq 5$) $\rightarrow$ Minor Repair; Fair ($5 <$ Rating $\leq 7$) $\rightarrow$ Major Repair.
- **Prioritization:** allocate budget Worst-First by sorting bridges by Health (ascending) and funding actions until the budget is exhausted.

**Significance:** Because this policy is derived from the same empirical dynamics used for matrix-based evaluation, it is a strong matrix-aligned heuristic reference while still respecting the hard budget. MPC-Oracle is kept separate because it uses privileged evaluator-aligned reward information during planning.

## 4.2. Evaluation Metrics

We evaluate each policy along 3 axes that matter in real-world IAM: *utility* (does the network get healthier?), *feasibility* (does the policy respect hard budgets?), and *behavioral fidelity* (how far does it deviate from historical practice?).

**Empirical matrix dynamics for evaluation.** To isolate decision quality from model-learning confounders, we run a matrix-based evaluation where next-year health transitions are sampled from an empirical one-year transition matrix estimated from the full dataset (Appendix C.4). This matrix serves as the evaluation environment's dynamics, capturing historical deterioration and the average efficacy of each intervention.

**Health Improvement.** Let $h_t^i$ be the health rating of bridge $i$ at year $t$. We report the average one-step health change per bridge, $\Delta H \triangleq \mathbb{E}[h_{t+1}^i - h_t^i]$ under the *executed* actions after budget enforcement. We present this as two relative scores: *Improve vs None* (compared to the No Action policy) and *Improve vs History* (compared to replaying the logged historical actions), matching the columns in Table 3.

**Budget Ratio.** We measure budget utilization as BudgetRatio $\triangleq \mathbb{E}\left[\frac{C(\mathbf{s}_t, \mathbf{a}_t)}{B_t}\right]$, where $\mathbf{a}_t$ is the executed joint

*Table 2.* Algorithm Feature Comparison

| Algorithm | Core Features | | | Advanced/Specific Features | | |
|---|---|---|---|---|---|---|
| | Explicit Constraint Handling | Single-Agent | Multi-Agent | Centralized Training (MARL) | Behavior Cloning | Value-Based |
| *Single-Agent Algorithms* | | | | | | |
| MultiTask-CPQ | ✔ | ✔ | ✗ | N/A | ✗ | ✔ |
| CDT | ✔ | ✔ | ✗ | N/A | ✔ | ✗ |
| OneStep | ✗ | ✔ | ✗ | ✗ | ✗ | ✔ |
| CQL | ✗ | ✔ | ✗ | ✗ | ✗ | ✔ |
| MPC+Forecasting | ✗ | ✔ | ✗ | N/A | ✗ | ✔ |
| MPC-Oracle | ✗ | ✔ | ✗ | N/A | ✗ | ✔ |
| MultiTask-BC | ✔ | ✔ | ✗ | N/A | ✔ | ✗ |
| *Multi-Agent Algorithms* | | | | | | |
| Random | ✗ | ✗ | ✔ | N/A | ✗ | ✗ |
| Discrete-BC | ✗ | ✗ | ✔ | ✗ | ✔ | ✗ |
| IQL-CQL-MARL | ✗ | ✗ | ✔ | ✗ | ✗ | ✔ |
| QMIX-CQL | ✗ | ✗ | ✔ | ✔ | ✗ | ✔ |
| QMIX-CQL-MF | ✗ | ✗ | ✔ | ✔ | ✗ | ✔ |

action (after allocation) and $B_t$ is the evaluation budget. Values close to 1 indicate full utilization of available resources.

**Behavioral Similarity.** To quantify behavioral shift, we compute Sim $\triangleq \mathbb{E}[\mathbb{I}(a_t^i = a_{t,\mathrm{hist}}^i)]$, i.e., the fraction of per-bridge decisions that match the logged historical action. This metric is informative in our domain because the dataset is extremely sparse in active interventions; therefore even small drops in similarity can correspond to substantial changes in *maintenance* frequency.

**Violation Rate (raw).** Since we enforce feasibility with the centralized greedy allocator (Section 3.2), we separately report the propensity of a *raw* policy to overspend. At timestep $t$, let $\mathbf{a}_t^{\mathrm{raw}}$ be the policy output. The raw violation indicator is $\mathbb{I}[C(\mathbf{s}_t, \mathbf{a}_t^{\mathrm{raw}}) > B_t]$, and *Violation Rate (raw)* is its empirical average over time and episodes. By construction, executed actions have zero violations.

**Health Gain per \$1M.** We report a scale-normalized cost-efficiency score HG/\$1M $\triangleq 10^6 \times \frac{\Delta H}{\mathbb{E}[C(\mathbf{s}_t, \mathbf{a}_t)]}$, where $\Delta H$ is the average health improvement per bridge and the denominator is the average executed cost per bridge. Higher values indicate better health gain per dollar.

**FQE Value (offline).** Complementing the matrix-based rollouts, we report an offline value estimate using Fitted Q Evaluation (FQE) (Le et al., 2019) on the test buffer. Let $r_t$ denote the per-step reward (aggregating health changes and penalties) defined identically to the training environment. FQE learns an action-value function $Q_\phi(s, a)$ to estimate the initial-state value as $V_{\mathrm{FQE}}^\pi \triangleq \mathbb{E}_{s_0 \sim d_0}[\sum_{t=0}^{T-1} \gamma^t r_t \mid a_t \sim \pi(\cdot \mid s_t, B_t)]$, where $d_0$ is the empirical distribution of initial bridge states. We report the mean and standard deviation of $V_{\mathrm{FQE}}^\pi$ across episodes; larger values indicate better long-horizon health outcomes under the training reward scheme.

## 5. Results and Analysis

We evaluate the algorithms around four research questions: (1) How do RL algorithms perform? (2) How sensitive are the results to allocation rules? (3) How do learned policies compare to heuristics? (4) How are actions distributed? (5) How do policies generalize over a 100-year horizon?

### 5.1. How do RL algorithms perform?

Table 3 summarizes the performance of single-agent and multi-agent algorithms. The normalized metrics are scaled so that CQL-Heuristic under the knapsack allocator equals 1.0. MPC-Oracle is included as an informed planning reference rather than a standard fair learning baseline, because it uses evaluator-aligned reward information during planning.

**Single-Agent Dominance.** Single-agent algorithms significantly outperform multi-agent reinforcement learning (MARL) approaches. Among learned RL methods in Table 3, CQL and MultiTask-CPQ achieve strong normalized improvements ($0.430 \pm 0.015$ and $0.394 \pm 0.423$, respectively), while MultiTask-CPQ has the best FQE score ($-0.0939$). Notably, MultiTask-CPQ's raw policy output maintains a low violation rate (1.7%) even before any external budget enforcement, demonstrating that the algorithm successfully internalized the cost constraints during training; CQL also keeps near-zero violations (0.7%).

**Planning-Based Methods.** MPC+Forecasting substantially outperforms imitation-style methods and reaches performance comparable to strong RL baselines, while remaining below MPC-Oracle, highlighting the importance of forecast quality in planning-based infrastructure management.

**The MARL Struggle.** Multi-agent algorithms exhibit po-

*Table 3.* Algorithm Performance Comparison (Updated with Mean ± SD). Normalized metrics are scaled so that the CQL-Heuristic result under the knapsack allocator is set to 1.0.

| Algorithm | Budget Ratio | Improve vs History | Behavioral Similarity | Violation Rate | Health Gain per $1M | FQE-score |
|---|---|---|---|---|---|---|
| *Single-Agent Algorithms* | | | | | | |
| MultiTask-BC | 0.824 ± 0.005 | -0.022 ± 0.001 | 0.990 ± 0.000 | 0.000 ± 0.000 | 0.352 ± 0.003 | -0.104 ± 0.010 |
| CDT | 0.199 ± 0.013 | -0.313 ± 0.010 | 0.989 ± 0.000 | 0.004 ± 0.000 | 0.642 ± 0.014 | N/A |
| CQL | 0.891 ± 0.006 | 0.430 ± 0.015 | 0.984 ± 0.000 | 0.007 ± 0.000 | 0.377 ± 0.008 | -0.104 ± 0.012 |
| OneStep | 0.963 ± 0.001 | -0.287 ± 0.143 | 0.036 ± 0.020 | 0.962 ± 0.020 | 0.123 ± 0.035 | -0.137 ± 0.034 |
| MultiTask-CPQ | 0.920 ± 0.007 | 0.394 ± 0.423 | 0.975 ± 0.000 | 0.017 ± 0.000 | 0.391 ± 0.004 | -0.094 ± 0.013 |
| *Multi-Agent Algorithms* | | | | | | |
| IQL-CQL-MARL | 0.829 ± 0.005 | -0.012 ± 0.000 | 0.997 ± 0.000 | 0.007 ± 0.000 | 0.367 ± 0.002 | -0.111 ± 0.009 |
| Discrete-BC | 0.821 ± 0.015 | -0.013 ± 0.004 | 0.997 ± 0.000 | 0.007 ± 0.000 | 0.370 ± 0.005 | -0.106 ± 0.009 |
| QMIX-CQL | 0.907 ± 0.077 | -0.067 ± 0.305 | 0.286 ± 0.150 | 0.711 ± 0.150 | 0.222 ± 0.132 | -0.209 ± 0.031 |
| QMIX-CQL-MF | 1.002 ± 0.002 | 0.085 ± 0.138 | 0.284 ± 0.182 | 0.715 ± 0.182 | 0.499 ± 0.055 | -0.183 ± 0.040 |
| Random-MARL | 0.965 ± 0.000 | -0.402 ± 0.000 | 0.250 ± 0.000 | 0.539 ± 0.000 | 0.079 ± 0.000 | -0.291 ± 0.024 |
| *Non-RL Methods* | | | | | | |
| MPC+Forecasting | 1.007 ± 0.001 | 0.417 ± 0.155 | N/A | N/A | 0.484 ± 0.071 | N/A |
| MPC-Oracle | 1.006 ± 0.000 | 1.185 ± 0.029 | N/A | N/A | 0.840 ± 0.014 | N/A |

larized performance, likely due to the scalability challenge posed by hundreds of agents (bridges). They either degenerate into pure imitation (e.g., IQL-CQL-MARL, Discrete-BC with $> 0.99$ similarity) resulting in negative improvements, or fail to coordinate entirely (e.g., QMIX-CQL), leading to chaotic behavior and catastrophic violation rates (71.1%).

QMIX-CQL-MF shows more stable behavior than standard QMIX-CQL, improving normalized improvement from $-0.067$ to $0.085$ while maintaining similar behavioral similarity and violation patterns. This suggests that compact mean-field aggregation can partially alleviate coordination challenges in large-scale bridge network management.

**Behavioral Fidelity and Action Sparsity.** High behavioral similarity scores ($> 0.97$) across top performers are expected, as the historical dataset is heavily dominated by "No Action" (Action 0) entries ($> 95\%$). Consequently, a deviation of $1.6\% \sim 2.7\%$ (as seen in CQL and CPQ) is statistically significant, representing a major shift in the distribution of *active* maintenance interventions. To facilitate learning in this sparse-reward setting, we employed a balanced sampling strategy during training, oversampling rare maintenance events to prevent the policy from collapsing into a trivial "always do nothing" solution. The success of MultiTask-CPQ lies in its ability to retain the necessary sparsity of interventions while optimizing the selection of the few active repair actions.

### 5.2. How sensitive are the results to allocation rules?

The final execution outcome depends not only on the learned scores or Q-values, but also on the budget-constrained allo-

cator that converts those scores into feasible interventions. We therefore evaluate whether the main conclusions are robust to alternative allocation rules rather than artifacts of a single post-processing choice.

*Table 4.* Allocator ablation under three allocation rules. Table values report normalized Improve vs History.

| Method | Greedy | Cost Ratio Greedy | Knapsack |
|---|---|---|---|
| CQL | 0.430 ± 0.015 | 0.662 ± 0.050 | 0.703 ± 0.072 |
| MultiTask-BC | -0.042 ± 0.002 | -0.027 ± 0.001 | -0.021 ± 0.001 |
| OneStep | -0.287 ± 0.143 | -0.137 ± 0.221 | -0.111 ± 0.233 |
| MultiTask-CPQ | 0.394 ± 0.423 | 0.599 ± 0.398 | 0.661 ± 0.431 |
| IQL-CQL-MARL | -0.014 ± 0.000 | -0.018 ± 0.001 | -0.009 ± 0.001 |
| QMIX-CQL | -0.067 ± 0.305 | 0.210 ± 0.529 | 0.265 ± 0.568 |
| QMIX-CQL-MF | 0.085 ± 0.138 | 0.406 ± 0.089 | 0.444 ± 0.119 |
| CQL-Heuristic | 0.708 ± 0.085 | 0.928 ± 0.038 | 1.000 ± 0.030 |

Allocator choice materially affects absolute performance: Cost Ratio Greedy and knapsack allocation generally outperform plain score-greedy allocation. However, the relative ranking of methods remains broadly consistent across allocators, indicating that the benchmark conclusions are not artifacts of a particular post-processing rule.

Methods with stronger value estimation or planning capability, such as CQL-Heuristic and QMIX-CQL-MF, benefit more from improved allocation strategies, while conservative imitation-style approaches (e.g., MultiTask-BC and IQL-CQL-MARL) remain relatively insensitive to allocator choice because their outputs already stay close to historical action distributions.

## 5.3. How do heuristics compare to RL?

We compare three approaches: a pure rule-based baseline ("Worst-First" heuristic), a pure offline RL baseline (CQL), and a hybrid heuristic-guided RL agent (CQL-Heuristic).

Here, CQL-Heuristic denotes a hybrid agent trained with the standard CQL objective plus an auxiliary imitation-style loss toward heuristic recommendations when available; inference uses the learned policy alone under the same budget allocator.

*Table 5.* Comparison of Heuristic and Heuristic-Guided RL Methods. Matrix Eval values follow the same normalization: CQL-Heuristic under the knapsack allocator is 1.0.

| Method | Type | Matrix Eval | FQE-score |
|---|---|---|---|
| Heuristic | Rule-based | **1.0176** | $-0.0998 \pm 0.0110$ |
| CQL | RL | 0.4302 | $-0.1043 \pm 0.0116$ |
| CQL-Heuristic | RL (Hybrid) | 0.7082 | **$-0.0911 \pm 0.0143$** |

Table 5 highlights the synergy between expert rules and data-driven learning.

**Matrix-aligned heuristic.** The Heuristic method remains the strongest rule-based reference in Matrix Evaluation (1.0176), as expected, since its rules are derived directly from the evaluation dynamics. We describe it as a heuristic reference rather than an oracle; MPC-Oracle is the separate planning reference that uses privileged evaluator-aligned reward information.

**Hybrid Superiority.** Crucially, the hybrid CQL-Heuristic agent significantly outperforms pure CQL in Matrix Evaluation (0.7082 vs. 0.4302), confirming that domain knowledge acts as a strong prior. Furthermore, it achieves the best overall FQE score ($-0.0911$), surpassing both pure RL and the heuristic baseline. This indicates the hybrid agent successfully combines the structural safety of rules with the optimization capability of RL, improving upon the heuristics rather than merely memorizing them.

## 5.4. How are actions distributed?

Table 6 compares each policy's *raw* output ("Original") against the *executed* actions after budget enforcement ("Constrained"), revealing how constraint handling reshapes the realized action distribution into distinct patterns. **Chaotic Explorers (OneStep, QMIX)** over-propose costly interventions, causing the allocator to frequently truncate requests; thus, their constrained distribution largely reflects budget clipping rather than a coherent strategy. In contrast, **Proactive Optimizers (CQL, CPQ)** propose targeted actions while remaining close to the feasible budget frontier, allowing the allocator to preserve their highest-value repairs. Finally, **Other clusters** show limited improvement: Imitators (BC/IQL) merely match the historical mix, while CDT

*Table 6.* Action distribution (%) under original and constrained settings. The gray Dataset row shows the historical action distribution. Larger values are shown with darker blue cells.

| Method | Original | | | | Constrained | | | |
|---|---|---|---|---|---|---|---|---|
| | **0** | **1** | **2** | **3** | **0** | **1** | **2** | **3** |
| Dataset | 99.08 | 0.41 | 0.42 | 0.10 | 99.08 | 0.41 | 0.42 | 0.10 |
| CQL | 98.59 | 0.50 | 0.83 | 0.08 | 99.02 | 0.39 | 0.53 | 0.06 |
| CQL-Heuristic | 97.61 | 1.36 | 0.96 | 0.07 | 98.89 | 0.57 | 0.49 | 0.04 |
| OneStep | 15.46 | 18.93 | 39.51 | 26.10 | 98.84 | 0.75 | 0.11 | 0.30 |
| OneStep-Heuristic | 17.04 | 9.31 | 46.45 | 27.21 | 99.07 | 0.37 | 0.14 | 0.42 |
| Single-BC | 99.11 | 0.31 | 0.51 | 0.07 | 99.13 | 0.30 | 0.50 | 0.07 |
| Random | 24.98 | 25.00 | 25.00 | 25.01 | 99.20 | 0.16 | 0.15 | 0.49 |
| CPQ | 98.17 | 0.76 | 0.95 | 0.12 | 98.97 | 0.45 | 0.48 | 0.09 |
| QMIX-CQL | 14.40 | 5.15 | 53.54 | 26.91 | 99.14 | 0.09 | 0.61 | 0.16 |
| IQL-CQL-MARL | 99.09 | 0.33 | 0.57 | 0.01 | 99.10 | 0.33 | 0.55 | 0.01 |
| Multi-BC | 99.09 | 0.34 | 0.53 | 0.04 | 99.09 | 0.34 | 0.53 | 0.04 |

collapses toward inaction.

## 5.5. How does RL perform in the long term?

While static evaluation provides a snapshot of performance, critical questions remain: How do these policies perform over an extended operational horizon, and how robust are they under distribution shift? To answer these, we designed a 100-year longitudinal simulation.

**Experimental Setup.** We initialize the simulation from the **static test states** so that $t = 0$ matches the in-distribution regime, then roll forward for a century, inducing an occupancy shift relative to the offline training distribution even though the transition mechanism remains data-derived. Additional setup details are provided in Appendix D.4.

**Long-term Performance Comparison.** We analyze cumulative cost and final average health after 100 years (Figure 2), revealing three representative behaviors. **Budget-Feasible Value Learning (CQL, CPQ):** CQL and CPQ achieve higher final health (3.04–3.29) than Random ($\approx 2.60$) while remaining within budget, indicating that offline value learning effectively prioritizes maintenance spending. **High Performance (Heuristic-Guided):** CQL-Heuristic attains the highest final health (3.41) at higher cost, consistent with a proactive, risk-aware maintenance strategy. **Inefficient Random or Myopic Repair:** Random expends resources through unstructured interventions, while OneStep spends aggressively without improving health, illustrating the limitations of random repair and myopic policy improvement under long-horizon OOD drift.

**Budget Sensitivity Analysis.** We investigate economic scalability by varying the global budget factor ($bf$) from $0.25\times$ to $4.0\times$ across six representative algorithms (Figure 3). The results reveal three distinct behavioral archetypes regarding resource utilization. **1) The Imitation Ceiling:** Pure imitation (Discrete-BC) fails to scale. Constrained by historical distributions, it cannot conceive of spending beyond historical norms; expenditure plateaus at $\approx \$182k$ even when

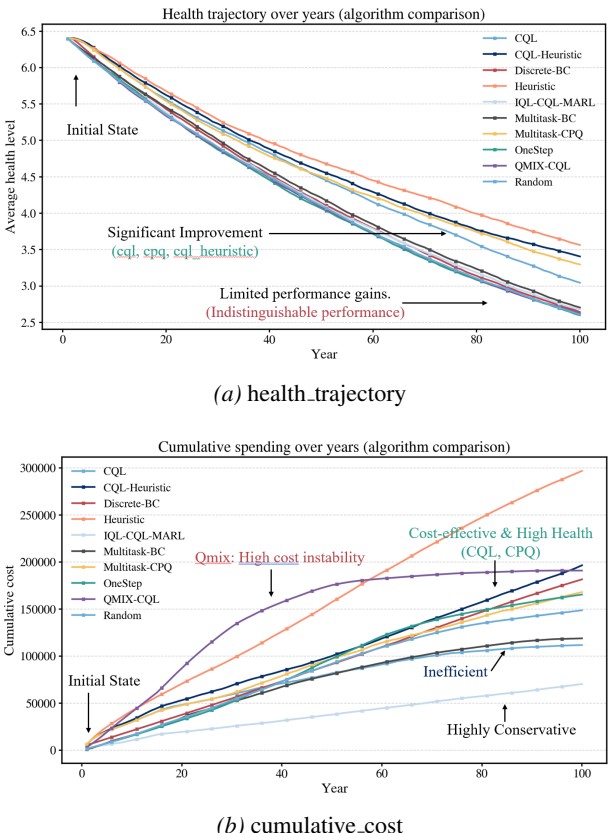

*(a)* health_trajectory

*(b)* cumulative_cost

*Figure 2.* Comparison of 100-year simulations. (a) health_trajectory (b) cumulative_cost

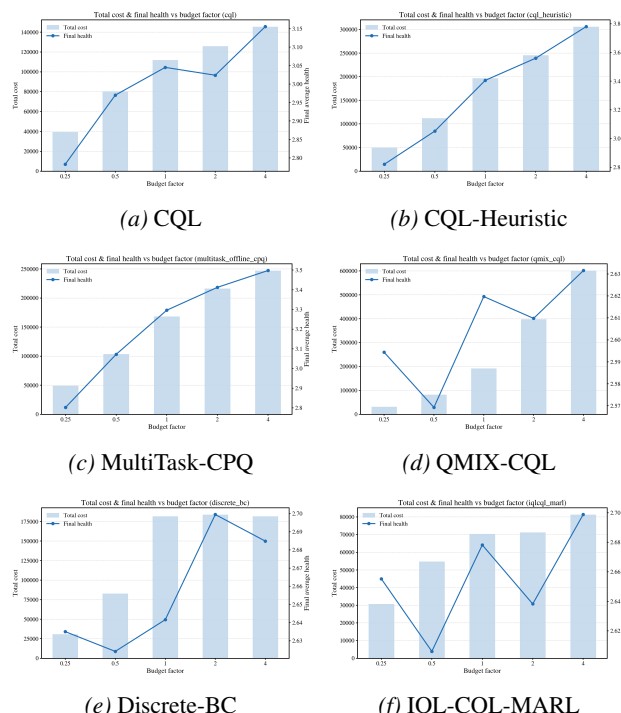

*(a)* CQL

*(b)* CQL-Heuristic

*(c)* MultiTask-CPQ

*(d)* QMIX-CQL

*(e)* Discrete-BC

*(f)* IQL-CQL-MARL

*Figure 3.* Budget scaling sensitivity analysis (3 rows × 2 cols). Each plot shows the algorithm's response (Health, Cost, Efficiency) to budget multipliers (0.25× to 4.0×) over 100 years.

$bf = 4.0$, leaving health outcomes stagnant. **2) Scalable Value Realization:** In contrast, value-based methods (e.g., CQL-Heuristic, MultiTask-CPQ) effectively convert surplus budget into long-term health. Notably, CQL-Heuristic demonstrates the most robust scalability, monotonically improving final health from 3.41 to 3.78 as resources grow. **3) Unstable Consumption:** Finally, QMIX-CQL exhibits "blind spending," where expenditure scales linearly with the limit, yet health outcomes remain erratic due to coordination failure. This comparison confirms that while imitation is limited by history, heuristic-guided value estimation possesses the necessary *economic scalability* to optimize under varying constraints. Appendix D.3 further separates allocator-side and agent-side budget scaling, showing both improve long-horizon outcomes.

## 6. Discussion and Conclusion

**InfraRL establishes a rigorous frontier** for offline constrained reinforcement learning. Our evaluation highlights three fundamental insights. First, regarding **Constraint Internalization**, specialized agents (e.g., MultiTask-CPQ) successfully internalize limits with a raw violation rate of only 1.7%. This proves that soft penalty mechanisms can approximate hard constraints, though external allocators remain necessary for absolute safety. Second, on **Economic Scalability**, we identify a critical limitation in imitation learning, which fails to improve with budget. In contrast, value-based methods (e.g., CQL-Heuristic) capture causal dynamics ($Cost \rightarrow Health$) rather than static correlations, allowing them to linearly scale performance with resources (4×) and generalize beyond historical data. Finally, **Inductive Bias** proves critical: 100-year simulations show pure RL degrades under shift, whereas heuristic-guided agents remain stable. We conclude that future IAM solutions require hybrid architectures that reconcile domain heuristics with data-driven optimization (see Appendix F).

InfraRL should be interpreted as a benchmark for offline policy optimization rather than a deployment-ready autonomous maintenance system. Real deployment would still face an offline-to-real gap arising from dynamics mismatch, occupancy shift beyond logged support, and administrative-to-operational factors such as contracting delays, traffic disruption, and human approval. Practical adoption would likely require rolling recalibration with newly observed records, conservative or uncertainty-aware learning, deterministic safety layers, and human-in-the-loop oversight.

More broadly, InfraRL is currently a single high-fidelity benchmark instance rather than a full benchmark suite. Future work includes richer constraint formulations and broader evaluation settings.

## Acknowledgements

This work is supported by the NSFC (62406270) and the STCSM Shanghai Rising-Star Program (24YF2748800).

## Impact Statement

This work introduces a benchmark for offline constrained decision-making in infrastructure asset management. Its potential positive impact lies in enabling more rigorous evaluation of resource-allocation methods for safety-critical public infrastructure. At the same time, the benchmark should not be interpreted as a deployment-ready decision system. Policies trained on historical administrative data may inherit historical biases, may not generalize to changing environmental or operational conditions, and should be used only with domain oversight and human approval. We discuss these limitations and the offline-to-real deployment gap in Section 6 and Appendix F.4.

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

## Appendix Contents

## A. Related Work

Our work sits at the intersection of infrastructure management, offline reinforcement learning, and constrained optimization. This section reviews the literature in these domains and situates our contribution within the existing landscape.

**Reinforcement Learning for Infrastructure Maintenance** The optimization of infrastructure maintenance is a classic sequential decision-making problem. While classical methods like dynamic programming are theoretically sound, they suffer from the "curse of dimensionality" when applied to large-scale networks. This has motivated a shift towards Reinforcement Learning (RL). Recent studies have successfully applied DRL to this domain, with some framing the problem centrally and others adopting a decentralized multi-agent (MARL) perspective. However, a critical barrier remains: these works predominantly rely on *online* learning simulators. They assume an agent can freely explore the state space—including allowing assets to fail—to learn optimal policies. This assumption is untenable for high-stakes public infrastructure. Our work addresses this by strictly adhering to an *offline* setting, learning solely from historical data.

**Domain-Specific Simulation Environments** Several platforms address infrastructure and sustainability but operate primarily

as online simulators. **IMP-MARL** (Leroy et al., 2023) is a prominent suite for infrastructure management planning (e.g., offshore wind farms). It supports up to 100 agents and models deterioration using physics-based simulations (Dec-POMDPs). While it effectively benchmarks online MARL algorithms like QMIX against heuristics, it relies on generated deterioration curves rather than historical inspection logs. Similarly, **SustainGym** (Yeh et al., 2023) provides environments for sustainable energy systems (e.g., EV charging) with realistic distribution shifts. These platforms differ from our work in two fundamental ways. First, they are designed for *online* interaction, whereas InfraRL provides a static dataset for *offline* learning where exploration is prohibited. Second, InfraRL focuses on the unique challenge of allocating a finite, renewable budget over decades to combat non-stationary degradation based on real-world administrative data.

**Benchmarks for Offline and Constrained RL** To situate InfraRL within the broader RL community, we compare it against influential benchmarks. As shown in Table 1, existing platforms lack at least one of the critical dimensions required for our target problem. **D4RL** (Fu et al., 2020) is the cornerstone of offline RL research but lacks native cost signals. **Safety-Gym** (Ray et al., 2019) focuses on safety hazards for online agents rather than budgetary constraints. **DSRL** (Ji et al., 2023) provides constrained offline datasets but focuses on robotic control tasks. InfraRL is the first to integrate all these dimensions: it is *offline*, involves *hard budgetary constraints*, supports *multi-agent* formulations, and is derived from *real-world* noisy administrative data.

# B. Detailed Algorithm Descriptions

This appendix provides detailed descriptions of the offline reinforcement learning algorithms evaluated in our benchmark. Each description outlines the core mechanism, architecture, and training objective of the respective algorithm.

## B.1. Multi-Agent Offline Algorithms

Our benchmark evaluates several multi-agent offline reinforcement learning algorithms. These methods are designed to learn decentralized policies from pre-collected, static datasets without further interaction with the environment, which is crucial for real-world applications where online exploration is infeasible. For the multi-agent algorithm, we refer to the implementation in og-marl (Formanek et al., 2023).

**Random.** The random baseline serves as a fundamental lower bound for multi-agent performance. In this setup, each agent independently and randomly selects an action from the available space at each step. This policy represents a complete lack of coordination or learned behavior. A specialized version can incorporate rudimentary budget awareness by prioritizing "affordable" actions, but it remains a non-learning-based approach.

**Discrete-BC.** This algorithm is a multi-agent adaptation of Behavior Cloning (BC) for discrete action spaces. It learns a decentralized policy by directly mimicking expert actions from the offline dataset. Each agent's policy is represented by a `DeepRNN` (Recurrent Neural Network), which processes the agent's local observation. To enable agents to distinguish themselves, a one-hot encoded agent ID is concatenated to each agent's observation. The training objective is to minimize the cross-entropy loss between the predicted action probabilities and the expert's chosen actions. This algorithm serves as a strong imitation learning baseline.

**IQL-CQL-MARL.** The IQL-CQL-MARL algorithm extends the Individual Q-learning (IQL) framework with Conservative Q-Learning (CQL) for multi-agent settings. Each agent learns its own Q-function using a `DeepRNN`, where agent IDs are also appended to observations. The core idea of IQL is to train each agent's Q-function independently. CQL is integrated to address the overestimation bias inherent in offline Q-learning by adding a regularization term to the objective that penalizes Q-values for out-of-distribution actions. The overall loss combines the standard TD (Temporal Difference) error with the CQL regularization term.

**QMIX-CQL.** QMIX-CQL combines the centralized training with decentralized execution (CTDE) framework of QMIX with CQL for offline learning. Individual agents learn their own Q-functions (using `DeepRNNs` with agent ID concatenation), but their Q-values are combined by a monotonic mixing network (`QMixer`) to produce a global Q-value. For offline learning, CQL regularization is applied to the mixed global Q-values, pushing down the values of actions not present in the dataset. This approach leverages multi-agent coordination through the mixer while ensuring conservative Q-value estimates for reliable offline policy learning.

## B.2. Constrained Single-Agent Offline Algorithms

To provide a contrasting perspective to the decentralized multi-agent approaches, we also evaluate several single-agent algorithms. These methods treat the entire system as a centralized control problem, where a single policy makes all decisions. This allows us to benchmark the performance of global optimization strategies, particularly those designed to handle explicit constraints. For the single-agent algorithm, we refer to the implementation in osrl (Liu et al., 2023a).

**MultiTask-BC.** This is a single-agent Behavior Cloning (BC) approach that learns a policy by mimicking expert demonstrations from the entire dataset. It is tailored for discrete action spaces and handles dynamic budget information by concatenating the current `budget` with the `observation` as input to the actor network (`MLPActorDiscrete`). This allows the policy to condition its actions on the remaining budget. Training is performed by minimizing the `nn.CrossEntropyLoss`.

**CQL (Conservative Q-Learning).** CQL is a widely used model-free offline RL algorithm designed to mitigate the value overestimation problem caused by the distributional shift between the dataset and the learned policy. It learns a Q-function by minimizing the standard Bellman error while simultaneously adding a regularization term that penalizes Q-values for actions not present in the dataset (out-of-distribution actions). By strictly regularizing the value function to remain close to the data distribution, CQL provides a robust policy that avoids risky, unknown actions. In our benchmark, it serves as a strong value-based baseline to compare against constraint-specific architectures.

**OneStep.** The OneStep algorithm represents a simplified, low-variance approach to offline learning designed to avoid the error propagation issues common in iterative dynamic programming. Instead of training a policy through recursive bootstrapping, it performs a single step of policy improvement over the behavior policy (approximated via behavior cloning). The algorithm estimates the Q-values (and potentially Cost Q-values) of the behavior policy and updates the target policy to maximize the expected return subject to constraints locally. This results in a highly stable policy that stays close to the demonstrated behavior while attempting to improve performance.

**CDT (Constrained Decision Transformer).** CDT adapts the Decision Transformer architecture for constrained environments. It is designed to strictly adhere to constraints by conditioning its predictions on both future returns (return-to-go) and future costs (cost-to-go). The transformer predicts actions based on the trajectory context (past states, actions, and rewards), while the attention mechanism allows it to respect cumulative cost constraints. The model is trained to imitate expert trajectories from the dataset that satisfy the given budget constraints.

**MultiTask-CPQ (Constrained Policy Q-learning).** MultiTask-CPQ is an adaptation of Constrained Policy Q-learning (CPQ), a model-free, off-policy algorithm for constrained RL with discrete action spaces. This "budget-aware" algorithm maintains two Q-networks: one for the expected cumulative reward (`q_net`) and another for the expected cumulative cost (`qc_net`). Both networks take the concatenated `state` and `budget` as input. For constraint handling, a cost threshold (`qc_thres`) is calculated based on the cost limit. Actions predicted to lead to a future cumulative cost exceeding this threshold are pruned by setting their Q-values to a very low number, ensuring the policy learns to avoid constraint violations.

## B.3. Inference-Time Constraint Handling: Centralized Greedy Allocator

While the algorithms described in Sections B.1 and B.2 focus on learning policies from offline data, strictly adhering to hard budget constraints during evaluation requires a dedicated execution mechanism. As introduced in Section 2, we employ a **Centralized Greedy Allocator** as a post-processing wrapper during the inference phase. This ensures that all evaluated baselines—regardless of their internal constraint handling capabilities—are compared under identical hard constraints.

**Mechanism Rationale.** The budget allocation problem at each timestep $t$ is treated as a variant of the *Knapsack Problem*, where the "value" of an item is the agent's estimated utility (Q-value or logit) and the "weight" is the maintenance cost. To resolve violations where the raw joint action $\mathbf{a}_t^{raw}$ exceeds the evaluation budget $B_{\text{eval}}$, we adopt a greedy approach based on the agents' confidence.

**Operational Logic.** Let $\mathcal{N}$ be the set of agents in a region. At timestep $t$, the allocator processes the raw actions as follows:

1. **Identify Active Requests:** Identify the subset of agents $\mathcal{A}_{active}$ requesting non-zero cost actions.

2. **Score Aggregation:** For each agent $i \in \mathcal{A}_{active}$, a priority score $S_i$ is retrieved from the model's output:

   - For **Value-based methods** (e.g., CQL, CPQ, OneStep): $S_i = Q(s_t^i, a_i^{raw})$.
   - For **Policy-based methods** (e.g., BC, CDT): $S_i$ is the unnormalized logit or probability probability associated

with the chosen action $a_i^{raw}$.

3. **Sorting:** Agents in $\mathcal{A}_{active}$ are sorted in descending order of their scores $S_i$.

4. **Greedy Allocation:** The allocator iterates through the sorted list. For the highest-priority agent, the action is approved if its cost is less than or equal to the remaining budget ($B_{remain}$). If approved, $B_{remain}$ is updated. If the budget is insufficient, the agent's action is forced to $a = 0$ (No Action).

5. **Fallback:** Any agent not in $\mathcal{A}_{active}$ or denied funding defaults to Action 0.

This deterministic mechanism guarantees that $\sum \text{Cost}(a_t^i) \leq B_{\text{eval}}$ for every step of the evaluation, prioritizing the actions where the agents have the highest estimated value or confidence.

## B.4. Worst-First Heuristic (Matrix-Aligned)

This heuristic baseline matches the one used in the main paper ("Worst-First" / matrix-aligned heuristic). It is a fully deterministic, rule-based policy: **the per-state action is fixed *before* evaluation** using the empirical one-year transition dynamics, and **during rollout the policy only performs table lookup + worst-first budget allocation**. MPC-Oracle, introduced below, is kept separate because it uses privileged evaluator-aligned reward information.

**Offline pre-computation (once, before evaluation).** We construct a deterministic action table $\pi : \mathcal{S} \to \mathcal{A}$ for discrete health states using the empirical one-year transition matrix $\mathcal{T}(s' \mid s, a)$. For each health state $s$, we select the action that maximizes the expected next-step health score:

$$\pi(s) = \arg\max_{a \in \mathcal{A}} \sum_{s' \in \mathcal{S}} \mathcal{T}(s' \mid s, a) \cdot V(s'), \tag{5}$$

where $V(s')$ is the numeric condition score (higher is better). Crucially, this optimization is performed *offline* to populate the lookup table and is *not* re-solved online at every timestep.

**Online execution (during rollout).** At each year $t$ within a region:

1. **Worst-First sorting:** sort bridges by current health (ascending).

2. **Action lookup:** for each bridge $i$ in order, set $a_t^i \leftarrow \pi(s_t^i)$ by table lookup.

3. **Budget allocation:** execute actions greedily until the annual budget is exhausted; remaining bridges default to No Action.

This yields a strong, domain-plausible baseline that is aligned with the matrix dynamics used for evaluation while remaining strictly rule-based at execution time.

## B.5. Details of Additional Baselines

**MPC+Forecasting.** MPC+Forecasting (implementation: `forecast_mpc_clean.py`, class `ForecastMPCClean`) is a short-horizon MPC baseline that relies only on a learned one-step health predictor. It inherits from `ForecastMPCBase` and uses `HealthTransitionMLP` to predict the next normalized health from the current observation and a one-hot action. The imagined observation is then advanced by the base-class rollout logic: the predicted health is written back into the observation, age is incremented by one year, and other bookkeeping follows the shared forecast-MPC state update. For each candidate first action $a$, the method rolls out in parallel (i) the imagined trajectory starting with $a$ and (ii) a full-horizon noop trajectory. Over horizon $H$, it scores the candidate by the discounted cumulative predicted health advantage over noop,

$$\sum_{t=0}^{H-1} \gamma^t \left( h_t^{\text{path}(a)} - h_t^{\text{noop}} \right), \tag{6}$$

where health is computed after de-normalization. The imagined rollout is scored entirely by predicted health and does not use the evaluator's Markov transition matrix, making this a fair learned/planning-style baseline. The noop action is normalized by fixing $Q[:, 0] = 0$, so the knapsack stage uses $Q(s, a) - Q(s, 0) = Q(s, a)$ for $a > 0$.

**MPC-Oracle.** MPC-Oracle (implementation: `forecast_mpc_informed.py`, class `ForecastMPCInformed`) is a stronger informed planning reference with privileged evaluator-aligned reward information. It requires `set_transition_matrices(tm)` before evaluation or planning. At each imagined step, the method reads the current de-normalized health from the observation and, for each agent, computes the scalar immediate reward using `calculate_expected_health_improvement(health, action, tm)`. These rewards are accumulated with discounting. The selected first-step action is the candidate action $a$, while all subsequent actions in the imagined rollout are noop (action 0). The imagined state is still advanced using `forecast_next_norm_batch` followed by `advance_obs_after_prediction`. Thus, MPC-Oracle is not a strictly learning-only fair baseline; it is an informed planning reference because its rewards are aligned with the evaluator's transition matrix.

**QMIX-CQL-MF.** QMIX-CQL-MF (implementation: `QMIXCQL_meanfield.py`, class `QMIXCQLMultiAgent`) combines QMIX-style monotonic mixing, mean-field context, and CQL-style conservative regularization for variable-number-agent settings. The mean-field module computes masked group-level statistics over observations—per-dimension mean, standard deviation, minimum, maximum, and active ratio—via `compute_mean_field_raw_features`, then encodes them with an MLP `MeanFieldEncoder` into `mf_embed`. Per-agent Q-values are produced by a GRU-based `DeepRNN` that receives the local observation concatenated with the mean-field embedding and, optionally, an agent identifier; `train_step` computes sequence Q-values only over valid agents. For mixing, `SimplifiedQMixer` uses `mf_embed` as the global state, and a hypernetwork generates nonnegative mixing weights that combine per-agent $Q_i(a_i)$ into $Q_{tot}$, structurally analogous to QMIX but with the full joint state replaced by a compact mean-field embedding. The TD target uses the mask-weighted average team reward plus $\gamma$ times `target_mixer` applied to max-next per-agent Q-values and the next mean-field embedding. Training includes adaptive CQL regularization with logsumexp suppression of out-of-distribution actions, legal-action masking, auxiliary entropy/spread/min-Q style terms, a CQL weight that increases with `update_count`, and soft target-network updates including the mean-field encoder.

## C. Detailed NBI Data Processing Pipeline

Our dataset construction follows a comprehensive multi-stage pipeline designed to transform raw NBI records into a structured reinforcement learning benchmark. This is shown in Figure 4.

### C.1. Data Acquisition and Preprocessing

Our benchmark is derived from the National Bridge Inventory (NBI), a longitudinal database maintained by the U.S. Federal Highway Administration (FHWA) that contains comprehensive records for all public road bridges in the United States.[1] Our multi-stage data processing pipeline transforms these raw records into a structured reinforcement learning benchmark.

**Data Extraction and Cleaning.** We utilized annual NBI data files from 1992 to 2023, performing an initial filtering to retain only highway bridges ('SERVICE_ON_042A' = 1) located in California. This focus on a single state and infrastructure type minimizes confounding variables related to differing state policies and environmental conditions.

To address common issues in historical data and ensure logical consistency for the MDP formulation, we implemented a rigorous cleaning protocol. Specifically, we filtered out data points that violated physical or economic causality:

1. **Phantom Improvement:** Records showing an improvement in structural health rating without any recorded maintenance action.

2. **Ineffective Repair:** Records where a maintenance action was logged, but the structural health rating decreased in the subsequent year.

3. **Phantom Cost:** Records showing maintenance expenditures (Cost > 0) but labeled as "No Action".

4. **Administrative Redundancy:** Sequences of reported maintenance actions and cost estimates—often identical across consecutive years—that fail to yield any health improvement. We hypothesize that these entries represent administrative reporting or carry-overs rather than actual physical execution. To isolate genuinely executed actions, we applied a rule-based filter: a recorded action is considered valid and retained only if it coincides with a change in any structural condition rating or an update to the reconstruction year.

---

[1] The NBI public data is accessible at: https://www.fhwa.dot.gov/bridge/nbi/ascii.cfm

Missing temporal data, such as structural evaluation scores, were imputed using a forward-fill followed by a backward-fill strategy to maintain the temporal consistency of each bridge's condition history. To ensure that our analysis was based on assets with sufficient historical context, bridges with fewer than 20 years of records within our study period were excluded.

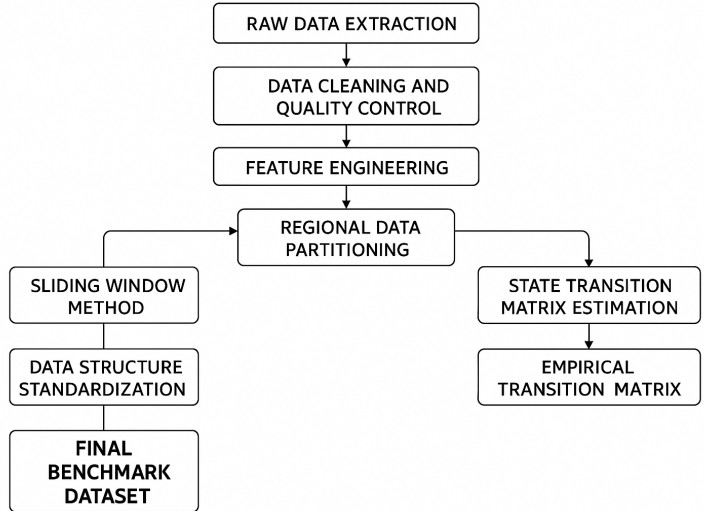

*Figure 4.* The pipeline for constructing the benchmark dataset from raw NBI data. The process includes data preprocessing, MDP formulation, regional partitioning, and yields the final dataset and an empirical transition matrix.

**Feature Engineering.** To enrich the state representation beyond raw NBI fields, we engineered several informative features designed to capture critical aspects of a bridge's condition and importance. These include 'bridge_age' (calculated as 'current_year' - 'YEAR_BUILT_027'), which is a primary driver of deterioration. Finally, we defined a composite 'importance_score' as a weighted sum of normalized ADT, age, and span length to approximate the bridge's systemic importance, helping an agent to prioritize critical assets.

**Action Space Definition and Cost Estimation.** To create a tractable action space, we abstracted the specific maintenance codes from the NBI field 'WORK_PROPOSED_075A' into four discrete, high-level actions: No Action, Minor Repair, Major Repair, and Replacement.

The mapping logic has been refined based on cost distribution analysis. Notably, code 33 (Deck Widening) is grouped with "No Action" (0). This decision stems from the fact that the average recorded cost for this activity was negligible ($\approx$\$33) and its occurrence frequency was extremely low, rendering it statistically insignificant as a standalone structural intervention. Code 31 is assigned to Minor Repair, while codes 34, 35, and 36 constitute Major Repair. The resulting dataset exhibits a distribution where the ratio of maintenance actions (Major:Minor:Replacement) is approximately **4:5:1**.

The cost associated with each action, 'cost(a_t)', was empirically derived by directly calculating the average of the 'TOTAL_IMP_COST_096' values across all corresponding historical interventions. Table 7 provides a detailed summary of this mapping and the resultant cost structure used throughout our experiments.

*Table 7.* Mapping of NBI Work Codes to Action Categories and Associated Costs

| Action Category | Assigned NBI Work Codes | Calculated Avg. Cost ($) |
|---|---|---|
| No Action | 0, 33 | 0.00 |
| Minor Repair | 31 | 1148.81 |
| Major Repair | 34, 35, 36 | 2317.70 |
| Replacement | 32, 37, 38 | 3004.33 |

### C.2. National Bridge Inventory (NBI) Feature Details

To provide clarity on the raw data used, this section details the key NBI items referenced in our pipeline. Definitions are based on the FHWA's *Recording and Coding Guide*. The bridge's health state is categorized into four distinct levels based on structural evaluation scores: **Good** (rating $\geq$ 7), **Fair** ($5 \leq$ rating $< 7$), **Poor** ($3 \leq$ rating $< 5$), and **Critical** (rating $< 3$).

**Condition Ratings (Items 58, 59, 60).** The core of the state representation is derived from the condition ratings for the Deck, Superstructure, and Substructure. Each is rated on a 0-9 scale, as detailed in Table 8. Our primary health score is the minimum of these three values, representing the weakest link principle.

*Table 8.* NBI Condition Rating Scale and Our Health State Categorization

| Code | Description | Our Categorization |
|------|-------------|--------------------|
| 9 | EXCELLENT | Good |
| 8 | VERY GOOD | Good |
| 7 | GOOD | Good |
| 6 | SATISFACTORY | Fair |
| 5 | FAIR | Fair |
| 4 | POOR | Poor |
| 3 | SERIOUS | Poor |
| 2 | CRITICAL | Critical |
| 1 | "IMMINENT" FAILURE | Critical |
| 0 | FAILED | Critical |

**Action and Cost Items.** Our action space and costs are derived from the following fields:

- `WORK_PROPOSED_075A`: This field indicates the type of work proposed to be done on the bridge. Our updated action mapping uses the following codes: 31 (Widening), 32 (Deck Replacement), 33 (Deck Widening), 34 (Rehabilitation), 35 (Repair), 36 (Strengthening), 37 (Painting), 38 (Other).

- `TOTAL_IMP_COST_096`: The estimated total cost of the improvement proposed in Item 75A, recorded in thousands of dollars. We use this to derive the empirical cost for our action space.

**State and Feature Items.** The following fields are used to construct the state space and engineered features:

- `SERVICE_ON_042A`: Type of service on the bridge. We filter for code '1', indicating a highway bridge.

- `YEAR_BUILT_027`: The year the bridge was originally constructed, used to calculate 'bridge_age'.

- `AVERAGE_DAILY_TRAFFIC_029`: The average number of vehicles per day carried by the bridge.

- `DECK_WIDTH_052`: The out-to-out width of the bridge deck in meters, used to calculate 'traffic_density'.

- `LAT_016` & `LONG_017`: The latitude and longitude of the bridge, used for our geographic regional partitioning.

### C.3. Reward Function Formulation

The reward function follows the main-paper definition in Section 2. For a single bridge (agent), we define the realized one-step reward as a sum of health improvement, a log-normalized cost penalty, and a potential-based reward shaping (PBRS) term:

$$r(s_t, a_t, s_{t+1}) = (h_{t+1} - h_t) - \beta \left[ \frac{\log(1 + \text{cost}(a_t))}{\log(1 + C_{95\%})} \right] + \tau(\gamma h_{t+1} - h_t), \tag{7}$$

where $h_t$ is the structural condition rating and $C_{95\%}$ is the 95th percentile of historical maintenance costs used for log normalization. The PBRS term is induced by the potential function $\Phi(s) = \tau h$, yielding $\gamma \Phi(s_{t+1}) - \Phi(s_t) = \tau(\gamma h_{t+1} - h_t)$.

**Parameter Selection.** We adopt a fixed set of parameters across experiments:

- **Cost weight ($\beta$):** 0.08

- **PBRS scale ($\tau$):** 0.15

- **Cost normalizer ($C_{95\%}$):** 5966.30

- **RL discount factor ($\gamma$):** 0.99

This configuration ensures that the agent is sufficiently penalized for expensive actions (like Replacement) while being strongly incentivized to maintain the bridge in a high health state.

### C.4. State Transition Matrix Estimation

We constructed an empirical state transition matrix, $P(s_{t+1}|s_t, a_t)$, for each of the four action types by aggregating all observed one-year transitions in the dataset and normalizing the counts into probabilities. These matrices (visualized in Figure 5) reveal distinct transition patterns across different condition states.

The matrix for No Action exhibits a strong diagonal and sub-diagonal structure, consistent with the natural tendency of bridges to remain in their current state or deteriorate over time.

For active interventions, the effectiveness varies by initial state. In the 'Poor' state, Minor Repair is the predominant action observed; transitions for Major Repair and Replacement were statistically insignificant due to limited sample size and are therefore not analyzed in detail for this state. Conversely, starting from a 'Good' state, Major Repair and Replacement show a significant impact in maintaining high condition ratings. Interestingly, in the 'Fair' state, Minor Repair demonstrates effectiveness comparable to Major Repair in preventing deterioration. This suggests that while our action categorization is based on cost intensity, the functional impact of lower-cost interventions (Minor Repair) remains substantial, particularly for bridges in intermediate conditions. Finally, the Replacement matrix generally concentrates probability mass in the 'Good' state, reflecting asset renewal.

### C.5. Regional Partitioning and Episode Generation

**Neighborhood-based Regional Partitioning.** To create meaningful multi-agent scenarios that reflect localized management challenges, we partitioned the statewide dataset into geographically coherent regions. Instead of a global clustering algorithm, we employed a neighborhood-based sampling method. To form a single region, a bridge was first randomly selected from the entire pool to act as a seed. Then, the $N-1$ bridges with the smallest Manhattan distance (calculated from their latitude and longitude coordinates) to this seed were identified. This collection of $N$ bridges—the seed and its nearest neighbors—constitutes one region. This process was repeated, sampling without replacement, until 400 distinct regions were generated. This approach ensures that agents within an episode manage a set of geographically proximate assets and face realistic, localized resource competition. For each region, a static, binary connectivity matrix $\mathbf{W}$ is constructed based on geographical proximity, capturing the inter-bridge spatial relationships.

**Sliding Window for Episode Generation.** From each region's full time-series data, we generated multiple episodes by applying a sliding window of 15 years with a stride of 5 years. This technique augments the number of distinct trajectories available for training while ensuring that each episode maintains its internal temporal coherence. This process resulted in a final benchmark dataset of 2000 episodes.

### C.6. Final Dataset Structure and Normalization

**Data Structure Specification.** Each episode in the final benchmark is stored as a dictionary-like object containing a collection of NumPy arrays. The dimensions below represent a single episode, with $T$ being the time horizon and $N$ the number of agents in the region. The arrays include: 'obs_arr' $[T, N, \text{obs\_dim}]$, 'act_arr' $[T, N]$, 'rew_arr' $[T, N]$, 'cost_arr' $[T, N]$, the static 'connectivity' matrix $[N, N]$, the shared 'budget_arr' $[T]$, and a 'metadata' dictionary containing episode-specific information.

**Global Normalization.** All continuous features were normalized using parameters computed solely from the training set to prevent data leakage. We employed a mixed strategy tailored to feature characteristics: **Z-score normalization** for features with Gaussian-like distributions (e.g., traffic density); **Min-max normalization** for features with defined bounds (e.g., bridge age); and **Robust scaling** (using median and interquartile range) for features with significant outliers, such as maintenance costs.

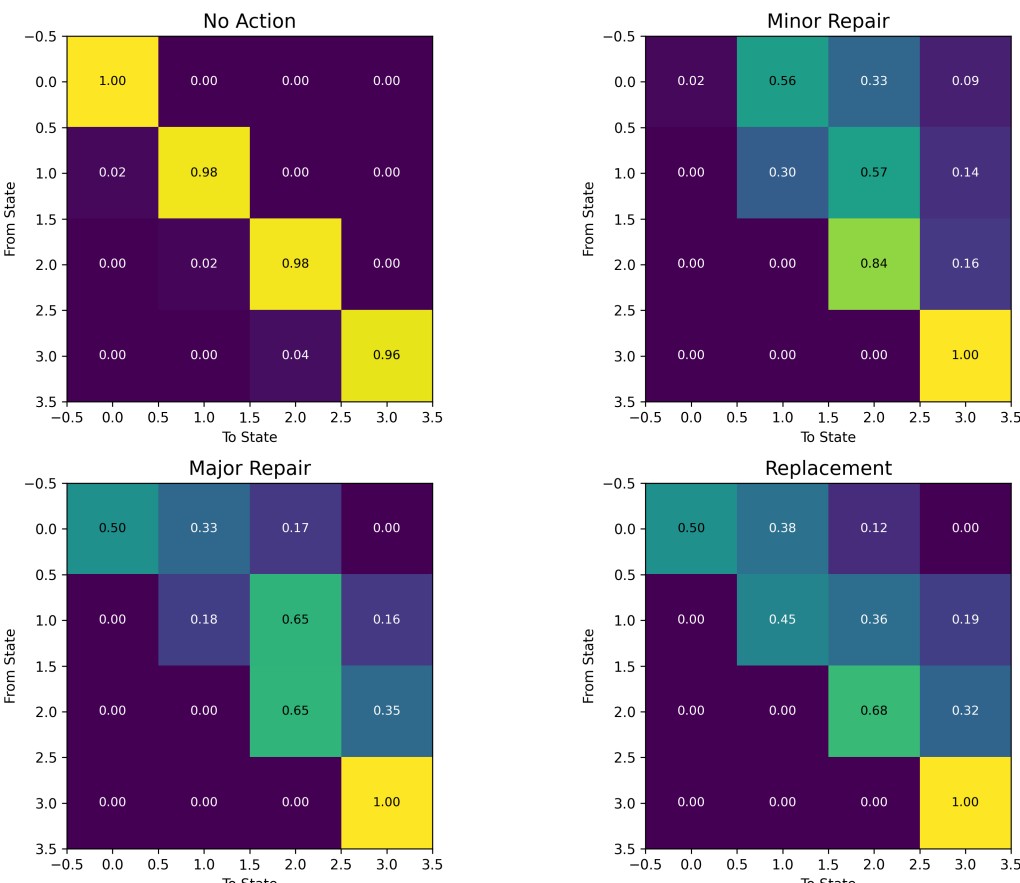

*Figure 5.* State transition matrices for different maintenance actions. Each matrix shows the probability of transitioning from one health state (rows) to another (columns) given a specific action. The four sub-figures correspond to: (top-left) No Action, (top-right) Minor Repair, (bottom-left) Major Repair, (bottom-right) Replacement.

## D. More Experimental Details

This section provides a more granular analysis of the experimental results presented in Table 3, offering deeper insights into algorithm performance, budget sensitivity, and the effectiveness of various resource allocation strategies.

### D.1. How is the algorithm performance?

Table 3 presents a comprehensive quantitative evaluation of the algorithms on the held-out test set. We analyze the performance dynamics based on health improvement, budget efficiency, and safety compliance.

**MultiTask-CPQ and CQL: The Value-Maximization Leaders.** Contrary to the assumption that constrained RL requires sacrificing performance for safety, MultiTask-CPQ emerges as one of the most effective learned agents. It achieves a strong normalized health improvement relative to historical baselines $(0.394 \pm 0.423)$ while maintaining a budget ratio of $0.920$, effectively utilizing available resources without overspending. Crucially, it demonstrates successful *constraint internalization*, keeping the violation rate at a negligible $1.7\%$ $(0.017)$. Similarly, CQL delivers strong positive gains $(0.430 \pm 0.015)$ with an even lower violation rate $(0.7\%)$. Both algorithms exhibit high behavioral similarity $(> 0.97)$ to human priors. As discussed in the Introduction, this high similarity does not imply a lack of novelty; rather, it reflects the algorithms' ability to navigate the extreme sparsity of the action space (where "No Action" dominates) while selectively optimizing maintenance interventions where they matter most.

**MultiTask-BC, Discrete-BC, and IQL-CQL-MARL: The Imitation Ceiling.** The imitation-based and conservative

multi-agent baselines illustrate the limits of purely mimicking historical data. While MultiTask-BC and Discrete-BC achieve near-perfect behavioral similarity (0.990 and 0.997) and zero violations, they fail to improve network health, resulting in negative normalized gains ($-0.022 \pm 0.001$ and $-0.013 \pm 0.004$, respectively). Their budget ratios ($\approx 0.82$) suggest an overly conservative strategy that "plays it safe" by withholding necessary maintenance funds. This confirms our finding on *economic scalability*: without a value function to drive optimization, imitation agents simply replicate historical sub-optimality and cannot leverage the full budget to improve infrastructure conditions.

**CDT: The Collapse to Inaction.** The Conditional Decision Transformer (CDT) presents a unique failure mode. Despite a high behavioral similarity of $0.989$, it incurs a large normalized health degradation ($-0.313 \pm 0.010$). The budget ratio of $0.199$ reveals the cause: the model has collapsed into a trivial solution of predicting "No Action" almost exclusively. Since the dataset is heavily skewed towards inaction, the model maximizes likelihood by doing nothing, ignoring the critical minority of samples that require maintenance.

**OneStep and QMIX-CQL: The Stability Failures.** Finally, OneStep and QMIX-CQL demonstrate the risks of unconstrained or poorly coordinated learning. OneStep (greedy planning) fails catastrophically with a $96.2\%$ violation rate and the worst health decline, proving that myopic optimization is infeasible for long-horizon infrastructure management. `Qmix_cql` similarly struggles, with a $71.1\%$ violation rate and low behavioral similarity ($0.286$), indicating that the joint value function factorization failed to stabilize in this offline setting, leading to erratic and unsafe policies.

## D.2. How is the algorithm's sensitivity to budget?

To investigate "Economic Scalability"—the ability of an agent to translate increased financial resources into tangible infrastructure health gains—we conducted a 100-year longitudinal simulation. We scaled the total budget from $0.25\times$ to $4.0\times$ the historical average. Figure 3 illustrates the response of six key algorithms, revealing a sharp dichotomy between value-based and imitation-based approaches.

**Value-Based Scalability (MultiTask-CPQ, CQL, CQL-Heuristic).** The algorithms relying on Q-value estimation (Figures a, b, and c) demonstrate strong budget awareness. As the budget limit (blue bars) increases, the network health (yellow line) shows a corresponding upward trend. This indicates that these agents correctly identify that a higher budget allows for more high-value maintenance actions. Notably, CQL-Heuristic outperforms the standard CQL. While CQL shows a performance dip at the $2.0\times$ budget level—suggesting potential instability or over-estimation in out-of-distribution high-budget states—CQL-Heuristic maintains a robust, monotonic increase in health. This suggests that combining learned Q-values with heuristic constraints provides the most stable mechanism for utilizing excess capital.

**The Imitation Ceiling (Discrete-BC).** In stark contrast, the imitation-based algorithms (Figures e) fail to scale. Despite the availability of $2\times$ or $4\times$ the budget, their actual expenditure remains practically flat, mirroring the $1\times$ historical baseline. Because these agents simply mimic the data distribution, they cannot conceive of a policy that spends more than the historical experts did, even when funds are unlimited. Consequently, their health performance plateaus immediately; they cannot turn extra money into better infrastructure, confirming their lack of economic scalability.

**Instability and Saturation (QMIX-CQL and General Trends).** `Qmix_cql` (Figure d) represents a failure mode of "blind spending." It is the only algorithm where expenditure scales linearly with the budget limit (using whatever is available), yet the health outcome is erratic and unstable. This confirms that without a coherent single-agent value signal, multi-agent coordination simply burns resources inefficiently. Finally, a general saturation trend is observed across most successful agents: expenditure grows rapidly up to the $1.0\times$ mark (doubling spending as budget doubles), but slows significantly beyond $1.0\times$. This implies a "maintenance saturation point"—once the critical backlog is addressed (around the historical budget level), the marginal utility of additional spending diminishes, as there are fewer high-priority repairs remaining.

## D.3. Agent-Side Scalability vs. Allocator Clipping

To disentangle whether budget-sensitivity gains arise from the agent itself or merely from reduced clipping by the post-hoc allocator, we added an ablation that separately varies the allocator budget factor ($a$) and the agent-perceived budget factor ($p$). Holding $p = 1$ while increasing $a$ relaxes only the allocator; increasing both to $(4, 4)$ also informs the agent that more resources are available. Table 9 shows that both changes improve long-horizon performance, indicating that the gains are not explained only by reduced clipping.

For both CQL and CPQ, increasing the allocator budget alone improves final health and score, confirming that reduced clipping matters. However, the additional improvement from $(4, 1)$ to $(4, 4)$ shows a separate agent-side scalability effect:

*Table 9.* Agent-side scalability vs. reduced clipping by the allocator. We separately vary allocator budget ($a$) and agent-perceived budget ($p$).

| Method | $(a, p)$ | Final Health | Score |
|--------|----------|--------------|-------|
| CQL | $(1, 1)$ | 3.49 | 6.41M |
| CQL | $(4, 1)$ | 3.53 | 7.60M |
| CQL | $(4, 4)$ | 3.73 | 8.90M |
| CPQ | $(1, 1)$ | 3.65 | 5.90M |
| CPQ | $(4, 1)$ | 3.84 | 7.11M |
| CPQ | $(4, 4)$ | 4.05 | 8.08M |

when the policy is conditioned on the larger available budget, it selects actions that better exploit the relaxed constraint.

### D.4. Long-horizon (100-year) simulation setup

This subsection details the implementation of our 100-year rollouts used to evaluate long-term performance under distribution shift. We focus on the simulation *setup* only; long-horizon results are reported in the main paper.

**Environment rollout (100-year health evolution).** **Initial state.** For each region (episode), we initialize the long-horizon simulation using the real observation at Year 1 in the dataset. Specifically, we de-normalize the standardized `policy_obs` using the training-set normalization statistics to recover the initial raw features for each bridge (health, age, ADT, length, and importance). We additionally use the Year-1 recorded health array as the initial health level to remain consistent with the original structural assessment.

**State reconstruction and re-normalization.** At year $t \in \{0, \ldots, 99\}$, we reconstruct the raw feature vector from the simulated health and age ($\texttt{sim\_health}_t$, $\texttt{sim\_age}_t$) and time-invariant covariates (ADT, length, importance). We then re-apply the training-set normalization to obtain the standardized policy input $s_t$.

**Annual health/age update.** Given the current health category and the executed maintenance action, we sample the next-year health category from a one-year empirical transition matrix (`transition_matrices`) learned from historical logs. Bridge age is updated deterministically by $+1$ per year. Thus, the entire 100-year evolution is driven by the data-derived empirical transition dynamics, capturing both natural deterioration and intervention effects.

**Budget construction and execution.** We reuse the same evaluation protocol as in the main paper: yearly budgets are derived from a behavior-cloning (Discrete-BC) baseline with a contingency margin and optional `budget_factor` scaling (see Section 3.2), and any overspending raw action requests are projected back to feasibility by the same centralized greedy allocator (see Appendix B.3).

## E. Hyperparameter and Computational Details

This section provides a comprehensive summary of the hyperparameter configurations, training procedures, and computational environment used for all experiments. The parameters were determined through a combination of standard practices in offline reinforcement learning literature and a limited grid search. **Table 10** outlines the general optimization, network, and batching settings that were applied across most algorithms. Following this, **Table 11** delves into the unique architectural and algorithmic parameters specific to each model family, such as those for Decision Transformers and Conservative Q-Learning. Finally, **Table 12** details the hardware and software stack used for the experiments, along with the resulting training performance metrics like duration and memory usage.

## F. More Discussion

This section expands upon the core findings presented in the main text, offering a deeper analysis of the algorithmic behaviors, the implications of the "Imitation Ceiling," and the structural limitations of the current benchmark.

### F.1. Revisiting the Safety-Performance Trade-off

A prevailing assumption in constrained reinforcement learning is the existence of a harsh trade-off: to achieve state-of-the-art utility, an agent must push boundaries, inevitably risking constraint violations. Our results with MultiTask-CPQ and CQL

*Table 10.* General Training Hyperparameters. These settings were applied across all applicable algorithms unless specified otherwise.

| Parameter | Value / Setting |
|---|---|
| *Optimization* | |
| Learning Rate (Policy/Actor) | 3e-4 |
| Learning Rate (Value/Critic) | 1e-3 |
| Optimizer | Adam |
| Adam Betas ($\beta_1, \beta_2$) | (0.9, 0.999) |
| Learning Rate Schedule | Cosine Annealing with Warm Restarts |
| Weight Decay | 1e-4 |
| Gradient Clipping Norm | 10.0 |
| *Network Architecture* | |
| Hidden Layer Dimensions | 64-128 |
| Network Depth | 2-3 hidden layers |
| Activation Function | ReLU |
| Dropout Rate | 0.1 |
| *Batching and Training Loop* | |
| Batch Size (Single-Agent) | 1024 |
| Batch Size (Multi-Agent) | 16 episodes |
| Training Steps | 200,000 |
| Evaluation Frequency | Every 2 epochs |
| Early Stopping Patience | 20 evaluations |

*Table 11.* Algorithm-Specific Hyperparameters.

| Algorithm Family | Parameter | Value / Setting |
|---|---|---|
| **Decision Transformer (CDT)** | Context Length | 10 timesteps |
| | Embedding Dimension | 128 |
| | Transformer Layers | 4 |
| | Attention Heads | 8 |
| | Return-to-Go Normalization | Per-episode z-score normalization |
| | Layer Normalization | Applied |
| **Conservative Q-Learning (CQL)** *(Used in IQL-CQL, QMIX-CQL, CPQ)* | CQL Regularization Weight ($\alpha$) | Tuned in {2.0, 3.0} |
| | Discount Factor ($\gamma$) | 0.95 |
| | Target Network Update Rate ($\tau$) | 0.005 (soft update) |
| | Target Update Frequency | Every 100 training steps |
| | Temperature Parameter ($\beta$) | 1.0 |
| **Recurrent Networks (RNN)** *(Used in MARL-BC, IQL-CQL, QMIX-CQL)* | RNN Type | GRU |
| | RNN Hidden State Dimension | 64 |
| | Sequence Sampling Length | 32 |
| | Data Augmentation | Time-shift augmentation |
| **QMIX-CQL** | Mixer Network Hidden Dimension | 32 |
| | Centralized Training Sampling | Episode-based |

challenge this binary view.

**Constraint Internalization.** In our experiments, MultiTask-CPQ achieved a strong normalized structural health improvement ($0.394 \pm 0.423$) while maintaining a raw violation rate of only $1.7\%$. This challenges the commonly assumed constraint–performance trade-off. Rather than overspending, effective offline constrained agents remain close to the empirical intervention sparsity (the heavy tail of "No Action") while making targeted deviations in a small subset of states where maintenance is most impactful. Overall, these results suggest that strong performance and tight safety can be achieved simultaneously when the policy accurately captures the underlying cost–value relationship.

*Table 12.* Computational Environment and Training Performance.

| Component | Specification / Value |
|---|---|
| *Hardware Specifications* | |
| GPU | $10 \times$ NVIDIA GeForce RTX 2080Ti (11GB VRAM) |
| System RAM | 256 GB |
| Storage | High-speed SSD |
| *Software Dependencies* | |
| Framework | PyTorch 2.4.1, CUDA 13.0 |
| Multi-Agent Library | Custom implementation based on PyMARL |
| Data Processing | Pandas, NumPy, SciPy |
| *Training Performance Metrics* | |
| Total Training Time (All Algs) | ~180 minutes |
|    - Single-Agent BC | 15-20 minutes |
|    - Multi-Agent BC | 25-30 minutes |
|    - CDT | 60-75 minutes |
|    - CPQ | 25-30 minutes |
|    - IQL-CQL | 10-15 minutes |
|    - QMIX-CQL | 20-25 minutes |
| Peak GPU Memory Usage | 8-10 GB per GPU for largest models |

**The Role of the Allocator.** It is important to acknowledge that while CPQ and CQL internalize constraints well, they are not perfect. The **Centralized Greedy Allocator** remains a critical component for deployment. It acts as a deterministic safety layer, translating probabilistic safety (low violation rates) into hard constraints (zero violations). The success of this hybrid approach (Learned Policy + Rule-Based Allocator) suggests that for high-stakes infrastructure, it can be practical to combine a learned policy with a deterministic enforcement layer to guarantee feasibility.

### F.2. The Mechanism of Economic Scalability

The divergence between imitation learning (BC) and value-based RL (CQL/CPQ) under increasing budgets (Figure 3) reveals a fundamental difference in how these algorithms represent knowledge.

**Correlations vs. Dynamics.** Imitation agents like Discrete-BC learn the conditional probability distribution of actions given states, $P(a|s)$, based on historical data. Since historical records reflect a specific, constrained budget reality, the agent learns to associate "expensive repairs" with "rare occurrences." Even when given infinite budget, the agent cannot break this learned correlation; it faces an **Imitation Ceiling**. In contrast, value-based agents learn the underlying transition dynamics and reward functions ($Cost \rightarrow \Delta Health$). This represents a causal understanding of the environment. Consequently, when the budget constraint is relaxed, the value function can identify previously unaffordable but high-value actions as viable. This **Economic Scalability** strengthens the case for deploying RL over BC: value-based agents can plan for resource levels that have not been observed historically.

### F.3. Parameter Sharing vs. Multi-Agent Coordination

Our benchmark reveals a performance gap between algorithms treating bridges as independent samples with shared parameters (Single-Agent baselines) and those attempting explicit multi-agent coordination (MARL baselines).

**The Stability of Independence.** The "Single-Agent" algorithms in our setup (e.g., MultiTask-CPQ) operate by learning a policy $\pi(a|s^i)$ that is shared across all bridges. This effectively treats every bridge in the dataset as an independent sample, vastly increasing the effective sample size and stabilizing the learning of the value function. The coordination is then handled *ex-post* by the Centralized Greedy Allocator. In contrast, MARL algorithms like QMIX-CQL attempt to learn a joint value function or mixing network to capture inter-agent dependencies implicitly. Our results show this is difficult in an offline setting; the noise in estimating joint values from static data leads to unstable policies ($> 70\%$ violation rate). This suggests that for large-scale infrastructure where physical interactions between assets are weak (unlike robot swarms), independent learning with centralized resource allocation is a more robust paradigm than fully coupled multi-agent learning.

### F.4. Limitations and Future Directions

Despite the rigorous construction of InfraRL, several limitations define the scope of our findings and point toward future research.

**Data Horizon and Bias.** The NBI dataset spans approximately 30 years. For infrastructure assets with lifecycles exceeding 75 years, this window captures only a fraction of the degradation process. Furthermore, the "ground truth" data is not optimal; it reflects historical decisions constrained by past policies, politics, and funding. While our RL agents outperform this baseline, they are still initialized and regularized against this potentially suboptimal distribution.

**Simplification of Logistics.** Our simulation abstracts away complex logistical constraints. In reality, maintenance is not instantaneous; it involves contracting delays, lane closures, and traffic impact studies. We model costs purely monetarily, but the "social cost" of traffic disruption is a major factor in real-world decision-making. Future iterations of InfraRL could incorporate multi-objective reward functions that balance structural health against user disruption.

**OOD Robustness.** The 100-year simulations highlighted that pure RL agents can drift into Out-Of-Distribution (OOD) states where their policies degrade. The success of the CQL-Heuristic hybrid agent indicates that domain knowledge is a necessary stabilizer. Future work should focus on formally integrating engineering physics into the RL loss function (Physics-Informed RL) to guarantee sensible behavior even in states far removed from the historical training data.

## G. Statement on the Use of AI Tools

In the preparation of this manuscript, we used an AI writing assistant (Gemini) for language polishing and proofreading. The core ideas, methodology, experiments, and analysis are the original work of the authors. All AI-assisted edits were manually reviewed by the authors, who take full responsibility for the final content.

## H. Reproducibility and Code Availability

### H.1. Reproducibility Measures

**Deterministic Operations:**

- Fixed random seeds (42 ,1024, 2026 for training)

- Deterministic GPU operations where computationally feasible

- Consistent data loading order across experiments

- Fixed initialization schemes for all neural networks

**Environment Control:**

- Conda environment specifications with exact version pinning

- Detailed documentation of hardware configurations

### H.2. Data and Model Availability

To facilitate the review process and ensure reproducibility, we provide the complete source code, preprocessed datasets, and training scripts in the following repository:

https://github.com/BriSky-2021/InfraRL

The repository contains:

- **Preprocessed benchmark datasets** with comprehensive metadata

- **Trained model checkpoints** for all evaluated algorithms

- **Complete experimental configurations** with hyperparameter specifications

- **Evaluation scripts** for reproducing all reported results

- **Visualization tools** for generating paper figures and custom analyses

The benchmark will be maintained as an open-source project with regular updates incorporating new NBI data releases and community contributions.

