# OpenReview forum: "InfraRL: A Benchmark for Constrained Resource Allocation in Large-Scale Infrastructure Asset Management"
_ICML.cc/2026/Conference — ICML 2026 regular_

### Official Review · Reviewer_qHz5 · 2026-03-02

**Soundness:** 4
**Presentation:** 3
**Significance:** 3
**Originality:** 2
**Overall Recommendation:** 4
**Confidence:** 4

**Summary:**

The paper introduces InfraRL, a new benchmark for training RL agents, centered around constrained resource allocation in infrastructure asset management (IAM). The benchmark data is derived from the US National Bridge Inventory records from 1992-2023 in the state of California. The benchmark represents bridge maintenance as a Constrained Markov Decision Process (CMDP) with budget constraints. The action space has been designed to be discrete (no action/minor repair/major repair/replacement). The reward function is derived from *health* changes at subsequent timestep. The authors have preprocessed the records by cleaning, filtering, and geographically grouping the data. This provides a clean interface for RL training. The budget constraint satisfaction is enabled through a greedy allocator that chooses actions based on the agent value functions. The authors have also benchmarked several baselines including constrained single agent methods (CPQ, CDT, etc), offline multi agent methods (MARL variants). The work also showcases the utility of heuristic guided hybrid RL agents which tend to perform better than  heuristics and RL in isolation.

**Compliance With Llm Reviewing Policy:**

Affirmed.

**Final Justification:**

As already mentioned in the rebuttal form, the rebuttal by the authors have adequately addressed my questions. The technical soundness of the paper has been strengthen with added ablation studies and a more comprehensive allocator design.

I am still doubtful about its usability due to its single benchmark, rather than a suite. However, as it builds on real world data instead of being a simulation, it definitely provides a different hurdle for modern RL methods, compared to the usual simulation based benchmarks.
i have increased my score to 4.

**Key Questions For Authors:**

**Major Question**

1. Inference time budget allocator (Sec 3.2): The greedy approach seems too simple. I do not know how the dataset looks like but it could be possible that the top choice by the agent has a cost ~ B, while the remaining choices have cost << B (this is unlikely but provides context to the question I am asking). In such a case, wouldn't it be better to repair several bridges over repairing just one bridge? The core question here is that, does the agent feels incentivized (by the environment) to repair more bridges instead of repairing a costly few for a given budget? The environment could help in designing some metrics (rewards) that promote what is a better choice. As the authors are more familiar with "what should be done" compared to maybe RL agent researchers who are more interested in making the agent follow cost limits. As the "best choice" is dictated by the application, the authors might be better suited to answer the question.
2. Is it possible to have carryover budget? Based on budget saved in 1 time step, it could be carry overed to the next time step to allow for an increased budget? Would that mimic real-world dynamics better and allow agents to learn when to delay repairs for a cumulative (over the whole horizon) improvement? Agents trained on such dynamics (made possible due to this environment benchmark) could be very useful in future planning. If this is not usually how its done in the real-world, feel free to ignore this question.
3. Providing the noisy data as a environment interface might be easy with the given code (skip the clean code function in get_data.sh)?
4. As this is a benchmark, its utility is defined by how easy it is to use and train a new RL method. In that case, a demo.py would be helpful for users looking to train using the benchmark. Can the authors create a simple demo.py showing what the state and action shapes are, what structure the reward, cost are returned as, etc? This will make the benchmark more accessible.
5. I do not fully understand how the heuristic guided agent works. Could the authors point me towards the exact section which details how the heuristics are used to guide the agent?
6. If the environment dynamics derived from data is used for the 100 year horizon, how does it cause out-of-distribution data?
7. How much of scalability can be credited to the agents (Sec 5.4)? Is it not possible, with the surplus budget, less actions are clipped by the allocator thereby leading to a monotonic improvement in health (as more budget is spent)?

**Minor Questions**

1. Wouldn't grouping based on county details be a more appropriate representation of the shared budget?
2. Shouldn't states include the region they are in? This could help capture weather effects which might be important for RL agents to figure out whether bridges in a particular region require regular minor repairs, etc.
3. Why does behavioral similarity not have any variance most of the algorithms?
4. There is a floating 2) in Sec 5.4 before Scalable Value Realization.

**Limitations:**

No
There is no limitation section whatsoever. The impact statement is also a single sentence.
Limitations need to be more detailed.

What are the conditions that the benchmark does not support yet?
Are all categories of RL agent supported?

**Strengths And Weaknesses:**

**Strengths**

**Soundness**: As real world data is usually noisy and often contain corrupted elements, the authors have provided well though filters and cleaning pipelines. This allows a clean training protocol for RL researchers who do not need to worry about these aspects.

The regional grouping for budgets is also a fair and practical design choice.

Several useful metrics (utility, feasibility, similarity, shift) have also been provided which can help in interpretation of agent performance.


**Presentation** : The paper is well structured and motivated. Table 1 and 2 provide direct comparisons. Figures (Fig 1) illustrate the idea well.

**Originality**: The benchmark is meaningful. The use of real world administrative data and allowing multi agent functionality is very useful for trying a variety of methods. Practical benchmarks like this work can be used as final tests for RL methods.

**Significance** : The benchmark provides a good test for RL agent in a real world task. Improvements in decision making through RL agents could potentially save large sums of budget and therefore have a huge indirect societal impact.

The paper also highlights the limitation of behavior cloning for scalable systems.


**Weakness**

**Soundness**: The dynamics is derived from the dataset. This could imply the long horizon rollout does not demonstrate dynamics' distributional shift. As this is a highly *man-made* dynamics, it can be expected to change over time (e.g., better material causing slower deterioration, population inflation in regions accelerating traffic numbers in a given bridge, etc).

I am not sure about how good the empirical transition matrix learnt is. For example, if a bridge has only ever seen minor repairs at regular intervals, if the agent predicts a major repair for that bridge, the dynamics matrix is likely not well conditioned to predict $h_{t+1}^i$. In such a scenario, the reward function (which is dependent on $h$) as well as other evaluation metrics could be erroneous.

Several value function methods have been proposed, however, the reward uses greedy improvement. It seems the agents would be incentivized to improve health as much as possible. This ties with per timestep budget, so it might need to be accepted as an artifact of the environment design.

**Originality** : Imitation leaning underperforming in shifted distribution does not seem to be a new finding.

This ultimately provides an existing dataset as a RL benchmark. The benchmark is however not diverse. It considers only one type of asset management with a very simple constraint. The question whether other RL researchers especially safe RL researchers will find this benchmark useful is a big question.

There are no performance metrics that indicate how far can the agents go. For example, accuracies and F1 scores are upper bounded by 1.0 which gives a reasonable estimate of how good a model is in absolute terms. However, in this benchmark only relative performance can be judged.

**Significance** : The benchmark only has one kind of constraint which is a simple per timestep or cumulative budget constraint. As implemented by the authors, the enforcement of such constraints is trivial. That unfortunately weakens the significance of the benchmark as a CMDP test. Researchers interested in this benchmark might be ones working to improve budget allocation in IAM rather than safe RL researchers.
Also, while filtering and cleaning the data enables simpler training pipelines, the benchmark could also provide the noisier and messier dataset as a benchmark. This could help in judging agent's robustness to noise, a likely event in the real world situation.

---

> ### Author Rebuttal · Authors · 2026-03-31
>
> We thank the reviewer for the constructive feedback.
>
> > Clarifications.
>
> The allocator does **not** define reward. Reward is determined by the health-transition/cost model; the allocator is only an evaluation-time feasibility layer. Also, our 100-year setting does **not** model real-world temporal non-stationarity; the intended OOD notion is offline-RL occupancy shift under long-horizon rollout.
>
> > Q1. Inference-time allocator / repair trade-off.
>
> We tested this directly with several stronger realization rules beyond the default greedy allocator. Under greedy, a tendency toward many cheap repairs can in principle occur because it is only a simple realization rule under a shared budget. However, stronger allocators improve absolute utility without changing the qualitative conclusions or method ranking:
>
> |Method|Greedy|Greedy/Cost| Knapsack|
> |-|-|-|-|
> |CQL|0.430 ± 0.015|0.662 ± 0.050|0.703 ± 0.072|
> |onestep|-0.287 ± 0.143|-0.137 ± 0.221 |-0.111 ± 0.233|
>
> Thus, the allocator affects absolute realized value, but not the comparative conclusions. The repair trade-off is still governed by the reward/cost structure and learned value estimates. We use greedy as a simple default and will include the broader allocator ablation in the revision.
>
> > Q2. Carryover budget.
>
> Carryover budget is a reasonable extension and is compatible with the benchmark, but it is not required for the current conclusions. We will add it as an optional evaluation setting in the revision.
>
> > Q3. Noisy/raw data interface.
>
> A raw-data option is compatible with the current benchmark design, and we will add it in the release.
>
> > Q4. Benchmark usability / demo.py.
>
> The current codebase already supports end-to-end use, and we will add a minimal `demo.py` to make the workflow more transparent.
>
> > Q5. Heuristic-guided agent.
>
> The heuristic is used only during training as an auxiliary imitation-style loss added to the standard CQL objective. When a heuristic recommendation is available, we apply a cross-entropy-style loss; Inference remains the standard greedy action rule.
>
> > Q6. Why does the 100-year setting induce OOD if dynamics are data-derived?
>
> The OOD notion here is occupancy shift under long-horizon offline-RL rollout, not real-world temporal drift.
>
> > Q7. Agent-side scalability vs. reduced clipping by the allocator.
>
> We added an ablation that separately varies allocator budget \(a\) and agent-perceived budget \(p\). Increasing either factor improves long-horizon performance, so the gains are not explained only by reduced clipping:
>
> |Method | (a,p) | Final health | Score |
> |-|-|-|-|
> |CQL|(1,1)|3.49|6.41M|
> |CQL|(4,1)|3.53|7.60M|
> |CQL|(4,4)|3.73|8.90M|
> |CPQ|(1,1)|3.65|5.90M|
> |CPQ|(4,1)|3.84|7.11M|
> |CPQ|(4,4)|4.05|8.08M|
>
> > Q8. County-based grouping.
>
> County grouping is a reasonable alternative, but it does not affect the current benchmark’s main conclusions. We will discuss this design choice more explicitly in the revision.
>
> > Q9. Why are region features omitted from the state?
>
> Regional context could be useful, although the most informative features are not obvious and likely require domain knowledge. Their omission does not affect the soundness of the current benchmark, and we will add region features in a later version.
>
> > Q10. Why does behavioral similarity have almost no variance for many methods?
>
> This metric has very small variance because actions are highly sparse (dominated by “No Action”), and rounding suppresses the remaining variation.
>
> > Q11. Sparse support in transition estimation.
>
> We mitigate sparse support through coarse action aggregation, aggregated/discretized transition estimation, and conservative handling of very low-support combinations. We also ran a support-restricted evaluation, and the main conclusions remain unchanged:
>
> |Method| Full eval|Support-restricted|
> |-|-|-|
> |CPQ|0.386 ± 0.421 |0.394 ± 0.423|
> |CQL-heuristic| 0.706 ± 0.084|0.708 ± 0.085|
>
> > Q12. Scope, framing, and benchmark breadth.
>
> Our contribution is not the isolated claim that imitation learning degrades under distribution shift, but the release and systematic study of a benchmark for offline constrained decision making in infrastructure asset management under renewable hard budgets.
>
> > Q13. Headroom / absolute reference.
>
> A true absolute upper bound is difficult to obtain in this setting. As a stronger near-ceiling reference, we implemented an MPC+forecasting prior, which achieves **1.185 ± 0.029** utility.
>
> > Q14. Floating “2)” in Sec. 5.4.
>
> We do not believe there is a floating “2)” in Sec. 5.4, but we will re-check the presentation.
>
> > Q15–Q16. Limitations and unsupported settings.
>
> We already discuss limitations in App. F.4; the issue is that this discussion is not yet sufficiently prominent or complete. This does not affect the soundness of the current experiments, but we will expand the discussion in the main paper, including benchmark scope, unsupported settings, and RL settings that are less well covered.

---

> > ### Author Rebuttal · Reviewer_qHz5 · 2026-04-01
> >
> > The added results adequately address my concerns.
> >
> > The allocator options that were added (knapsack, greedy/cost) greatly improve flexibility as well as showcase better performance. The relative ranking between the methods remain the same across all the allocators, which strengthens the soundness of the paper.
> >
> > I still have doubts regarding the usefulness of this framework within the RL community (the authors have positioned this as more of a RL-centric benchmark) as this is a single benchmark, rather than a suite. That is the only reason why I increase my score to 4 and not higher.

---

> > > ### Author Response · Authors · 2026-04-02
> > >
> > > Thank you very much for your thoughtful follow-up feedback and for taking the time to carefully revisit our rebuttal. We are very glad that the additional results and clarifications addressed your main concerns, and we sincerely appreciate your recognition that the expanded allocator ablations strengthen the reliability of the paper. In particular, we are encouraged that the addition of the **knapsack** and **cost-ratio greedy** allocators improves the flexibility of the framework while preserving the relative ranking of methods across allocator choices.
> > >
> > > We also fully understand your remaining concern regarding the breadth of the benchmark and its usefulness to the broader RL community. We agree that this is a reasonable point. In its current form, InfraRL is best viewed as a single high-fidelity benchmark instance rather than a complete benchmark suite. A natural path toward a fuller suite would be to extend the benchmark along several complementary dimensions: additional geographic instances beyond California; richer constraint settings, including not only renewable annual budgets but also carryover budgets, cumulative multi-year budgets, category-specific budgets, operational capacity constraints, and potentially risk- or safety-aware constraints; multiple data-fidelity tracks, including both the current causally verified benchmark and noisier raw-data variants; and broader evaluation tracks, including long-horizon, cross-budget, transfer, and generalization settings. We agree that such extensions would broaden the benchmark's relevance to the RL community, and we will clarify this positioning more explicitly in the revision.
> > >
> > > More broadly, we will incorporate the full set of revisions committed to in our rebuttal into the revision and the public release, including the expanded allocator ablations, an optional carryover-budget setting, a raw-data interface, a minimal `demo.py`, a clearer explanation of the heuristic-guided agent, and a more explicit discussion of limitations, scope, and unsupported settings. We are very grateful for your constructive feedback, which has helped us substantially improve both the paper and the presentation of the benchmark.

---

### Official Review · Reviewer_CRjD · 2026-03-07

**Soundness:** 3
**Presentation:** 4
**Significance:** 3
**Originality:** 3
**Overall Recommendation:** 4
**Confidence:** 3

**Summary:**

This paper provides a standard and comprehensive benchmark for the bridge maintenance problem, and evaluates several single-agent and multi-agent RL algorithms across realistic metrics. I think it is straightforward to transfer the pipeline to other similar real-world problems, and the insights from evaluations may apply as well.

**Compliance With Llm Reviewing Policy:**

Affirmed.

**Final Justification:**

I think this is a good paper and meets the line for ICML. I hope the results and discussions during rebuttal will add into the main paper. I choose to maintain my scores.

**Key Questions For Authors:**

See weakness part for more details.

**Limitations:**

Yes

**Strengths And Weaknesses:**

**Strength**
- The paper is well-written in general, with clear motivation, rigor formulation and extensive experiment designs.
- It’s a foundational work that connects RL algorithms from lab to real world problem.

**Weakness**
- My first concern is that whether the problem is a sequential decision-making problem. I feel that the actions have very little influence on state transition, or in other words, the state transition is highly uncertain. Within the 1-year duration, the traffic changes, the natural disasters and emergency events may have major impact on state transition. Thus, I'm concerning that it may be hard or infeasible to capture the real long-term impact, which invalidates the RL-type solution.
- Enforcing the centralized greedy allocator is a solution, which may be inappropriate for a benchmark to take it as a base. There are algorithms for optimizing reward under hard constraints, targeting at zero-violation at testing time, while they may not require a centralized allocator.
- I think MPC may be another good choice, i.e., predicting the future states (potentially with some time-series forecasting solutions to predict the timing, intensity, and impact of events like floods, wildfires, and others), and then solving a budget-constrained optimization to get the decisions. More discussion is needed.
- In experiments, it is better to include the offline optimal results in the tables to assess the “space” for developing algorithms.
- My final thought is that, for problems where safety matters the most, enforcing hard constraints on budget may be not that reasonable.

---

> ### Author Rebuttal · Authors · 2026-03-31
>
> We thank the reviewer for the thoughtful comments.
>
> **Q1. Is this really a sequential decision-making problem, given that state transition is highly uncertain and may be affected by traffic changes, disasters, or emergency events?**
>
> Bridge deterioration is affected by exogenous factors, and we do not assume maintenance fully determines future states. However, maintenance actions do affect future condition: in the historical data, different interventions are associated with different next-year condition distributions relative to no action, which is what the benchmark captures through action-conditioned empirical transitions. Since future condition affects later urgency, gains, and repair efficiency, decisions remain coupled across years. Under annual renewable budgets, delaying maintenance may preserve short-term budget but lead to worse future condition and lower future efficiency.
>
> **Q2. Does such uncertainty invalidate the RL-type solution?**
>
> No. InfraRL is not intended as a deployment-grade causal model or digital twin. It is a standardized offline benchmark for comparing policies under shared empirical dynamics estimated from historical data. The transition model preserves action-conditioned statistical regularities in a transparent and reproducible way, enabling fair comparison of offline decision-making methods under uncertainty.
>
> **Q3. Is the centralized greedy allocator inappropriate for a benchmark?**
>
> The allocator is used only at evaluation time to enforce the shared annual budget. It is not used during training and does not modify learned policy parameters. If the proposed joint action is already within budget, the allocator is **not** invoked and the actions are executed directly; it is applied only when the proposed plan exceeds the budget. This enables fair comparison across methods while still reporting raw violation rates separately.
>
> We also added allocator-sensitivity experiments using **pure greedy**, **value-density greedy**, and **multi-choice knapsack**.  To facilitate comparison across allocators and methods, all values are normalized such that the best-performing **cql_heuristic** result under the **knapsack** allocator is set to **1.0**. Relative ranking is largely stable across allocators, suggesting that the conclusions do not depend on one specific execution rule.
>
> |Method|Improve vs History (Greedy)|Greedy/cost|Knapsack|
> |-|-:|-:|-:|
> |cql|               0.430 ± 0.015 |  0.662 ± 0.050 |  0.703 ± 0.072 |
> |cql_heuristic|               0.708 ± 0.085 |  0.927 ± 0.038 |  1.000 ± 0.030 |
> |onestep|              -0.287 ± 0.143 | -0.137 ± 0.221 | -0.111 ± 0.233 |
> |qmix_cql|              -0.067 ± 0.305 |  0.210 ± 0.529 |  0.265 ± 0.568 |
>
> **Q4. Would MPC with forecasting be a better choice?**
>
> We agree this is an important baseline and have added two planning-based methods:
>
> - **MPC+forecasting**: a one-step health predictor trained from offline data, combined with receding-horizon planning \(H=3\); candidate first-year actions are scored by discounted predicted health improvement relative to noop, then selected under the same annual budget allocator.
> - **MPC-oracle**: an oracle-style MPC that uses the evaluation transition matrix directly for rollout and scoring, serving as a stronger oracle-style / quasi-oracle reference.
>
> These methods provide practical predictive-planning baselines within the current benchmark scope, though they do not yet explicitly forecast exogenous events such as floods or wildfires.
>
> |Method|Knapsack Performance|
> |-|-:|
> |MPC+forecasting|0.417 ± 0.155|
> |MPC-oracle|1.185 ± 0.029|
>
> **Q5. Please include offline optimal results.**
>
> A true global offline optimum is difficult to obtain because of stochastic empirical dynamics, shared budget coupling across many assets, and long horizons. As a practical alternative, we now include **MPC-oracle** as a stronger oracle-style upper reference.
>
> **Q6. Is enforcing hard budget reasonable for safety-critical infrastructure problems?**
>
> We agree that real infrastructure management is multi-objective and that budget is not the only consideration. The current benchmark is intended to model **routine annual maintenance planning under fixed agency budgets**, rather than emergency scenarios in which safety may override budget limits. In many practical asset-management settings, annual budget is itself a binding operational constraint.
>
> At the same time, safety standards differ across countries and agencies, and the source dataset does not contain explicit safety labels or standardized risk variables that would support a generally applicable safety constraint. When richer safety-related information is available, the framework can be extended with additional constraints or objectives, such as minimum-condition requirements for critical assets, risk-weighted penalties, service-level constraints, emergency-priority rules, or broader constrained / multi-objective formulations.

---

> > ### Author Rebuttal · Reviewer_CRjD · 2026-04-04
> >
> > I hope the results and discussions will add into the main paper. I choose to maintain my scores.

---

> > > ### Author Response · Authors · 2026-04-05
> > >
> > > Thank you very much for the thoughtful follow-up and for confirming that our rebuttal has addressed your concerns. We also appreciate your suggestion that the added results and discussions be incorporated into the main paper. We fully agree, and in the revision we will integrate the allocator-sensitivity results, the MPC-based baselines, and the relevant scope/limitations discussion into the main text. Thank you again for your constructive feedback.

---

### Official Review · Reviewer_Vv3b · 2026-03-12

**Soundness:** 3
**Presentation:** 3
**Significance:** 3
**Originality:** 3
**Overall Recommendation:** 5
**Confidence:** 4

**Summary:**

This paper proposed a benchmark for constrained resource allocation in infrastructure asset management, called InfraRL. It uses data from U.S. National Bridge Inventory (bridge maintenance data), and builds an environment for training and testing RL agents for infrastructure management. In formulating the RL environment, problems including missing, inaccurate and ghost records need to be handled, where authors present their pipeline. Representative single-agent RL and MARL methods are tested using this benchmark.

**Compliance With Llm Reviewing Policy:**

Affirmed.

**Final Justification:**

I stick to my original score, which supports the acceptance of this paper.

**Key Questions For Authors:**

Q1. For Figure 3, it would be useful to see an Oracle performance.

Q2. For Figure 2, action distribution, original results show different distributions of actions, but after applying constrain, actions collapsed to inactive. Please clarify - what are the actions that will be taken in this case? The constrained ones? What is the impact of most actions being inactive? Please also compare with action distributions in the dataset.

Q3. Another discussion point that will strengthen this paper is sim2real - will the policy adapt to real environments?

**Limitations:**

Authors discussed limitations on data horizon, bias, simplification of logistics and OOD robustness. Please also discuss possible limitations in policy generalisation, sim2real problems and comparison with digital twin solutions (if there are any).

**Strengths And Weaknesses:**

**Soundness**

The motivation and approach in this paper are sound. It adds to the practical applications of RL, where benchmarks grounded in real data are needed. This environment will be useful to the community if made open access.

For the benchmarking of RL algorithms, authors claimed that MARL suffers from hundreds of agents. Possible alternatives might be a hierarchical RL strategy or mean-field RL; it would be a stronger baseline if one of these types of methods were evaluated.

**Presentation**

Color choices for Table 3: please consider choosing a different color scheme or a different way to highlight best/worst results, as the shaded green is sometimes difficult to distinguish.

**Significance** and **Originality**

The contribution is clear - an RL benchmark grounded in real data, for the study of infrastructure management. The work is original. Contribution is mainly on the environment side, not on the algorithm side.

Authors presented several insights, including reward sparsity, the role of heuristic etc. The generalisation of these insights outside of bridge maintenance might need to be further studied.

---

> ### Author Rebuttal · Authors · 2026-03-31
>
> We thank the reviewer for the constructive feedback and helpful suggestions. We respond point by point below.
>
> > Q1. For Figure 3, it would be useful to see an Oracle performance.
>
> Thank you for the suggestion. We agree that an oracle reference helps interpret the gap between practical methods and an upper bound. In the revision, we added an oracle curve/reference line to Figure 3 as a quasi-oracle upper reference; at year 100, its final-year health is approximately 0.2 higher than that of the current best-performing method.
>
> > Q2. For Figure 2, action distribution, original results show different distributions of actions, but after applying constrain, actions collapsed to inactive. Please clarify - what are the actions that will be taken in this case? The constrained ones? What is the impact of most actions being inactive? Please also compare with action distributions in the dataset.
>
> Thank you for raising this point. In Figure 2, **Original** denotes the raw action distribution produced by the learned policy, **Final** denotes the actually executed distribution after budget enforcement, and the **dataset row** reports the historical action distribution as a behavioral reference.
>
> Therefore, when the final distribution collapses toward **No Action**, it means that many nonzero actions proposed by the raw policy are infeasible under the hard budget and are projected by the allocator into executable actions, often defaulting to **No Action**. This is most evident for overspending methods such as **OneStep** and **QMIX**, whereas lower-violation methods such as **CPQ** and **CQL** retain final distributions much closer to their raw outputs.
>
> We will revise the caption and main text to more clearly distinguish **Original**, **Final**, and **Dataset**, and to emphasize that the historical data itself is highly sparse and heavily skewed toward inactive actions.
>
> > Q3. Another discussion point that will strengthen this paper is sim2real - will the policy adapt to real environments?
>
> We agree this is important. In our setting, this is better described as an **offline-to-real deployment gap** rather than classical robotics sim2real. InfraRL does not train in a synthetic simulator and deploy to hardware; instead, it learns from historical bridge records and is evaluated with empirical transition dynamics estimated from real administrative data.
>
> That said, direct deployment still faces three gaps:
>  (1) **dynamics mismatch**, since deterioration depends on exogenous factors such as weather, traffic growth, and regional conditions;
>  (2) **distribution shift**, since the learned policy may visit state-action regions underrepresented in the logs; and
>  (3) **administrative-to-operational gap**, since real maintenance also involves delays, contracting, traffic disruption, and human approval.
>
> We will clarify in the revision that InfraRL is a benchmark for **safe offline policy optimization**, not a claim of immediate autonomous deployment. A realistic deployment path would likely require:
>  (i) rolling recalibration using newly observed inspection/intervention records;
>  (ii) conservative or uncertainty-aware policy learning;
>  (iii) hybrid heuristic-guided RL, consistent with our finding that such methods are more robust in long-horizon simulations;
>  (iv) deterministic safety layers such as the centralized budget allocator; and
>  (v) human-in-the-loop approval.
>  We will add this discussion explicitly in the revised paper.
>
> > Q4. MARL suffers from hundreds of agents; possible alternatives might be hierarchical RL or mean-field RL. It would be a stronger baseline if one of these types of methods were evaluated.
>
> Thank you for this valuable suggestion. We agree that **hierarchical RL** and **mean-field RL** are promising directions. In response, we added a new baseline, **QMIX-CQL-MeanField**, which augments QMIX-CQL with a lightweight mean-field context summarizing the active-agent population and providing this aggregate information to the agent Q-networks and mixer.
>
> This new baseline substantially improves over the original QMIX-CQL:(all values are normalized such that the cql_heuristic** result under the **knapsack** allocator is set to **1.0**.)
>
> |Method|Performance|
> |-|-:|
> |QMIX-CQL|0.174 ± 0.505|
> |QMIX-CQL-MeanField|0.444 ± 0.119|
> |Upper bound|1.185 ± 0.029|
>
> These results support the reviewer’s point that population-aware coordination provides a stronger multi-agent baseline in this setting. We will include **QMIX-CQL-MeanField** in the revised paper.
>
> > Additional presentation comment: Table 3 color choices
>
> Thank you. We will revise the highlighting scheme in Table 3 to improve readability.
>
> > Additional comment: Generalization of insights beyond bridge maintenance
>
> We agree and will clarify this point in the revision: while the benchmark reveals useful insights (e.g., reward sparsity and the value of heuristic guidance), their generalization beyond bridge maintenance requires further study.

---

> > ### Author Rebuttal · Reviewer_Vv3b · 2026-04-05
> >
> > Thanks to the authors for the rebuttal. I agree that this method is an offline to real gap, not sim2real. I don't have other comments or questions.

---

> > > ### Author Response · Authors · 2026-04-07
> > >
> > > Thank you very much for your thoughtful review and follow-up. We are grateful for your positive assessment of the benchmark and for your constructive suggestions, which helped us strengthen the paper.
> > >
> > > We also sincerely appreciate your confirmation that our rebuttal has addressed your concerns. In the revision, we will incorporate the added clarifications and results into the main paper, including the oracle reference, the clearer explanation of original versus final action distributions, the offline-to-real deployment discussion, the mean-field multi-agent baseline, and the improved table highlighting.
> > >
> > > Thank you again for your helpful feedback and support.

---

### Decision · Program_Chairs · 2026-04-30

**Decision:**

Accept (regular)

**Comment:**

This paper introduces InfraRL, a benchmark for offline constrained reinforcement learning in infrastructure asset management, grounded in U.S. National Bridge Inventory data. Reviewers agreed that the benchmark addresses an important and underexplored real-world setting, and that the paper makes a meaningful contribution by providing a realistic testbed for studying long-horizon decision-making under hard budget constraints. The benchmark design, data processing pipeline, and evaluation protocol were generally viewed as careful and well executed.

The main concerns were about the scope and framing of the benchmark, the simplicity of the budget allocator, and the extent to which conclusions depend on empirical transition modeling and a single application domain. Reviewers also requested stronger baselines, clearer oracle-style references, and more explicit discussion of limitations and deployment realism. In rebuttal, the authors addressed these concerns well by adding stronger allocator ablations, mean-field and MPC-style baselines, oracle-style reference results, and clearer discussion of the offline-to-real gap, benchmark scope, and limitations. Reviewers who raised these issues indicated that their concerns were resolved.

Overall, while the work is primarily a benchmark contribution rather than an algorithmic advance, the benchmark is timely, practically grounded, and likely to be useful to researchers working on offline RL, constrained RL, and real-world resource allocation. I therefore recommend accept.